# FlowKac: An Efficient Neural Fokker-Planck solver using Temporal Normalizing flows and the Feynman-Kac Formula

**Naoufal El Bekri**                                             *naoufal.elbekri@gmail.com*
*IMT Atlantique, Lab-STICC, UMR 6285, 29238, CNRS, Brest, France*
*Univ Brest, CNRS, UMR 6205, Laboratoire de Mathématiques de Bretagne Atlantique, France*

**Lucas Drumetz**                                             *lucas.drumetz@imt-atlantique.fr*
*IMT Atlantique, Lab-STICC, UMR 6285, 29238, CNRS, Brest, France*
*ODYSSEY Team-Project, INRIA Ifremer IMT-Atlantique, 35042, CNRS, Brest, France*

**Franck Vermet**                                             *franck.vermet@univ-brest.fr*
*Univ Brest, CNRS, UMR 6205, Laboratoire de Mathématiques de Bretagne Atlantique, France*

**Reviewed on OpenReview:** *https://openreview.net/forum?id=paeyQFa5or*

## Abstract

Solving the Fokker-Planck equation for high-dimensional complex dynamical systems remains a pivotal yet challenging task due to the intractability of analytical solutions and the limitations of traditional numerical methods. In this work, we present FlowKac, a novel approach that reformulates the Fokker-Planck equation using the Feynman-Kac formula, allowing to query the solution at a given point via the expected values of stochastic paths. A key innovation of FlowKac lies in its adaptive stochastic sampling scheme which significantly reduces the computational complexity while maintaining high accuracy. This sampling technique, coupled with a time-indexed normalizing flow, designed for capturing time-evolving probability densities, enables robust sampling of collocation points, resulting in a flexible and mesh-free solver. This formulation mitigates the curse of dimensionality and enhances computational efficiency and accuracy, which is particularly crucial for applications that inherently require dimensions beyond the conventional three. We validate the robustness and scalability of our method through various experiments on a range of stochastic differential equations, demonstrating significant improvements over existing techniques.

## 1 Introduction

The Fokker-Planck equation (FPE) (Risken & Frank, 1996) describes the time evolution of probability density functions associated with diffusion processes, playing a central role in modeling stochastic dynamical systems across diverse domains, including physics (Lucia & Gervino, 2015), finance (Sornette, 2001), engineering (Langtangen, 1991), and biology (Degond et al., 2020). The interest of the FPE lies in the fact that it represents an Eulerian view (i.e. location-based) of the process rather than the usual Lagrangian view, based on trajectories (Batchelor, 2000). However, except in very specific cases, analytical solutions to the FPE are out of reach for systems with high-dimensional or complex dynamics, necessitating efficient numerical methods.

Traditional numerical approaches, such as finite difference, finite element, and path integral methods, face the curse of dimensionality when applied to high-dimensional problems where the computational cost grows exponentially (Schuëller, 1997). Since these methods rely on discretizing the solution space using a mesh, their scalability is inherently limited, motivating the need for alternative, mesh-free approaches. Deep learning-based methods have emerged as promising alternatives in recent years, offering greater flexibility and scalability in high-dimensional settings (Raissi et al., 2019; Yin et al., 2023).

In this work, we introduce FlowKac[1], a novel generative model that combines the Feynman-Kac formula with normalizing flows to efficiently solve the Fokker-Planck equation (FPE). The key features and advantages of our approach are as follows:

- Similar to Beck et al. (2021b); Mandal & Apte (2024), our method reformulates the FPE using the Feynman-Kac formula which expresses the solution as an expectation over stochastic paths. This reformulation allows us to define an efficient regression loss function which is computed by sampling stochastic trajectories—one of the main computational challenges of these methods. We significantly improve this process through the stochastic sampling trick, which leverages properties of stochastic flows and Taylor expansions to reduce the computational burden. Importantly, this sampling algorithm can be extended to other Feynman-Kac-based approaches.

- By leveraging normalizing flows, a class of flexible deep generative models, our approach is particularly well-suited for handling probability densities, allowing to access a continuous representation of the solution. Once trained, FlowKac acts as a powerful sampler and provides density evaluations without requiring additional SDE discretization and enabling efficient generation of new samples from the learned probability distribution. This generative capability allows us, analogous to Gabrié et al. (2022), to dynamically refine the training set by inverting the trained flow. By generating samples that increasingly align with the true probability density, this adaptive sampling strategy improves both training efficiency and the accuracy of the estimated density.

- Unlike Mandal & Apte (2024), which relies on stationary densities to infer time-dependent solutions, our approach remains effective even for stochastic differential equations (SDEs) with degenerate or non-existent steady-state distributions, broadening its applicability to a wider class of problems.

- Finally, being a neural-based method, our approach can be scaled to higher dimensions.

More broadly, the Feynman-Kac framework extends beyond the Fokker-Planck equation, encompassing a wide class of parabolic partial differential equations (PDEs), including the heat equation, reaction-diffusion systems, and other Kolmogorov-type equations. This implies that our approach could be naturally extended to address a wide range of problems in stochastic dynamical systems, statistical physics, and beyond, where such PDEs play a central role.

The remainder of this paper is structured as follows: In Section 2, we review the relevant literature on deep learning-based methods for solving partial differential equations, with a focus on approaches for the Fokker-Planck equation. Section 3 introduces the problem formulation and provides an overview of traditional non-neural numerical methods. In Section 4, we present our proposed approach, detailing the reformulation of the FPE, the construction of the loss function, and the implementation of both the baseline and advanced versions of our method, which incorporates an efficient stochastic sampling algorithm. Section 5 contains a series of numerical experiments comparing our method to a state-of-the-art neural approach, followed by concluding remarks in Section 6.

## 2 Related work

The present section discusses the relevant literature on deep learning approaches to solving partial differential equations (PDEs), with a particular focus on methods applicable to the Fokker-Planck equation (FPE), and puts it in context with the proposed approach.

A broad class of neural methods for PDEs revolves around circumventing the curse of dimensionality and handling complex boundary conditions efficiently. Among these methods, the Deep Galerkin Method (DGM), introduced in Sirignano & Spiliopoulos (2018), extends the traditional Galerkin method by using deep neural networks to approximate solutions to a PDE as a linear combination of simpler functions. Similarly, the Deep Ritz Method (Yu et al., 2018) employs a variational formulation to approximate solutions for elliptic PDEs. Another approach is the Deep Splitting Method (Beck et al., 2021a), designed for parabolic PDEs.

---

[1]Code available at `https://github.com/naoufalEB/FlowKac`.

This method subdivides the time domain into smaller intervals where the PDE is approximately linear, and a deep-learning approximation is applied to each subinterval.

Backward Stochastic Differential Equations (BSDEs) have also been explored to solve PDEs (Han et al., 2017; 2018; Raissi, 2024). The method employs coupled forward-backward SDEs, discretizing them to approximate the solution of the associated PDE. This method is particularly effective in dealing with high-dimensional PDEs, successfully scaling to PDEs in hundreds of dimensions.

One of the most prominent approaches in recent literature involves Physics-Informed Neural Networks (PINNs), introduced in Raissi et al. (2019). PINNs solve PDEs by incorporating the governing equation into the loss function thereby minimizing the residual using a least-squares approach. PINNs have been applied to both time-dependent and steady-state PDEs. More recently, extensions of PINNs that leverage generative models, such as Normalizing Flows, have been explored (Feng & Zhou, 2022; Tang et al., 2021). Such an extension uses generative models to adaptively sample collocation points, gradually enriching the training set and improving convergence while reducing the computational cost.

Reformulating the Fokker-Planck equation using the Feynman-Kac formula has proven to be a powerful approach. Several works such as Beck et al. (2021b); Sabate Vidales et al. (2021) adopt this strategy by discretizing the underlying stochastic differential equation (SDE) and solving a minimization problem on the expected value of the corresponding stochastic process. Another related strategy, described in Mandal & Apte (2024) exploits the steady-state solution of the FPE and the Feynman-Kac formula to derive a general time-dependent solution.

A recent paradigm explores solving the Fokker-Planck equation under the transportation of measure framework, which gives rise to a probability flow equation (Boffi & Vanden-Eijnden, 2023). In this approach, the initial probability density is pushed forward through a velocity field to yield the time-dependent density solution. This method links to recent advances in generative learning, such as score-based diffusion models (Song et al., 2021) and flow matching techniques (Lipman et al., 2023), offering a novel perspective on modeling time-evolving probability densities.

For a broader overview of neural-based approximation for PDEs we refer the reader to the surveys Beck et al. (2023); Blechschmidt & Ernst (2021); E et al. (2021).

## 3 Problem statment

### 3.1 Fokker-Planck equation

We consider a filtered probability space $(\Omega, \mathcal{F}, P)$ and a time horizon $T$, and define a diffusion process $X = \{X_t\}_{t \in [0,T]}$ by the Itô Stochastic Differential Equation (SDE):

$$dX_t = \mu(X_t, t)dt + \sigma(X_t, t)dW_t, \quad t \in [0, T], \tag{1}$$

where $W = \{W_t\}_{t \in [0,T]}$ is the $m$-dimensional adapted standard Wiener process, $\mu : \mathbb{R}^d \times [0, T] \to \mathbb{R}^d$ the drift coefficient and $\sigma : \mathbb{R}^d \times [0, T] \to \mathbb{R}^{d \times m}$ the diffusion matrix. Both functions are time-dependent and assumed to satisfy global Lipschitz conditions ensuring the existence and uniqueness of solutions to the SDE (Stroock & Varadhan, 1997). Moreover, these regularity conditions guarantee the existence and smoothness of the associated probability density function (PDF), which evolves according to the Fokker-Planck equation:

$$\frac{\partial p}{\partial t} = \mathcal{F}p := -\nabla \cdot (\mu p) + \frac{1}{2} \nabla \cdot \left( \nabla \cdot \left( \sigma \sigma^\top p \right) \right), \tag{2}$$

where $\nabla \cdot$ denotes the divergence operator. Specifically, when applied to a matrix, $\nabla$ operates column-wise, meaning the divergence is computed separately for each column of the matrix.

To explicitly represent Equation 2, let $x = [x_1, x_2, ..., x_d]^\top$, $\mu = [\mu_1, \mu_2, ..., \mu_d]^\top$ and $D = \frac{1}{2} \sigma \sigma^\top$, where $D$ is the diffusion matrix, thus the FPE becomes:

$$\frac{\partial p}{\partial t}(x, t) = -\sum_{i=1}^{d} \frac{\partial}{\partial x_i} \left[ \mu_i(x, t) p(x, t) \right] + \sum_{i,j=1}^{d} \frac{\partial^2}{\partial x_i \partial x_j} \left[ D_{ij}(x, t) p(x, t) \right], \qquad x \in \mathbb{R}^d, \, t \in [0, T],$$

and the initial condition is specified as:

$$p(x, 0) = \psi(x), \quad x \in \mathbb{R}^d.$$

The solution is subject to the normalization and non-negativity constraints:

$$\int_{\mathbb{R}^d} p(x, t)dx = 1 \quad \text{and} \quad p(x, t) \geq 0, \quad \forall t \in [0, T], \tag{3}$$

and the additional boundedness condition:

$$p(x, t) \to 0 \text{ as } \|x\|_2 \to \infty, \tag{4}$$

ensuring $p$ is a valid probability density.

Over the years, significant work has been done to develop numerical solutions for the Fokker-Planck equation, spanning classical discretization schemes to modern neural network-based approaches. The following sections provide a concise overview of these approximation techniques.

## 3.2 Non-learning-based integration methods

Analytical solutions for the Fokker–Planck equation have been developed for only a limited number of low-dimensional systems. This limitation has led to a large body of approximation theory and techniques (Schuëller, 1997; Hundsdorfer & Verwer, 2003), each leveraging different perspectives to address the intractability of the exact solutions. These include:

1. **Classical density estimation techniques**: These methods approximate solutions through random sampling, estimating the probability density function empirically using techniques such as histograms or kernel density estimation (KDE).

2. **Path Integral (PI) Approach**: The PI method (Naess & Johnsen, 1991) approximates the probability density function as a Gaussian over an infinitesimal time interval, which is then integrated to obtain a global solution.

3. **Maximum Entropy Methods (MEM)**: In the MEM framework (Sobezyk & Trebicki, 1990; Sobczyk & Trebicki, 1992), the solution is approximated through the various moments and constrained to maximize the informational Shannon entropy.

4. **Grid-Based Numerical Solutions**: Grid-based techniques discretize the FPE and solve it through numerical integration and linear algebra (Pichler et al., 2013). These methods span a spectrum of formulations:

   - **Finite Differences (FD)**: FD leads to an explicit scheme where the values can be calculated directly, eliminating the need for matrix inversion. Despite its simplicity, explicit finite difference methods often face stability issues, necessitating implicit formulations for more reliable solutions.
   - **Alternating Direction Implicit (ADI)**: ADI is another significant and improved finite difference scheme. In ADI, finite difference steps in each direction are resolved separately, treating one dimension implicitly and the others explicitly in each step. This approach results in a stable finite difference formulation. The primary advantages of ADI include a tridiagonal matrix, which allows for efficient computation.
   - **Finite Element Methods (FEM)**: FEM partitions the domain into smaller subdomains, allowing flexible handling of complex geometries and boundary conditions.

Each numerical method offers unique advantages in terms of stability, efficiency, and applicability to complex geometries. However, the main drawback that all these methods suffer from is the curse of dimensionality arising when handling high dimensional FPE. As these methods rely mostly on a discretized grid, the computational complexity scales exponentially with the grid dimension, while the grid precision heavily impacts the approximation accuracy.

### 3.3 Neural network solutions: The Physics-informed approach

A notable paradigm that addresses the dimensionality constraint is machine learning, which offers mesh-free solutions. A particularly promising method for solving differential equations is the Physics-Informed Neural Network (PINN) approach. This technique integrates knowledge from physical laws defined by partial differential equations and boundary conditions. In the PINN framework, a neural network, parameterized by a set of weights $\theta$ and computing $p_\theta$, is employed to approximate $p^*$, the true solution to the Fokker-Planck equation (FPE). The neural network achieves this approximation by minimizing a combination of two loss functions: one related to the dynamics of the Fokker-Planck operator $(\mathcal{L}_\mathcal{F})$, where the derivatives of the neural-based probability density $p_\theta$ are efficiently computed through automatic differentiation, and another enforcing the initial condition $(\mathcal{L}_i)$. The training set consists of $N_\mathcal{F}$ spatial points and $M_\mathcal{F}$ temporal points, sampled uniformly from a given domain. The total loss function is then defined as:

$$\mathcal{L}(\theta) = \mathcal{L}_i(\theta) + \mathcal{L}_\mathcal{F}(\theta) \tag{5}$$

where

$$\mathcal{L}_\mathcal{F}(\theta) = \frac{1}{N_\mathcal{F} \times M_\mathcal{F}} \sum_{k=1}^{N_\mathcal{F}} \sum_{j=1}^{M_\mathcal{F}} \left( \frac{\partial p_\theta}{\partial t}(x^k, t^j) - \mathcal{F}p_\theta(x^k, t^j) \right)^2$$

and

$$\mathcal{L}_i(\theta) = \frac{1}{N_\mathcal{F}} \sum_{k=1}^{N_\mathcal{F}} \left( p_\theta(x^k, 0) - \psi(x^k) \right)^2$$

PINNs have demonstrated remarkable efficacy in modeling and approximating solutions to a wide range of differential equations, as comprehensively reviewed in Cuomo et al. (2022). However, despite their widespread success, recent work by Mandal & Apte (2024) reveals a fundamental limitation of the PINN framework when applied to solving the Fokker-Planck equation and deriving a time-dependent solution. Specifically, if we assume that Equation 2 has a strong, unique solution $p^*$, one can show that there exists a sequence of functions $(f_n)_{n \in \mathbb{N}}$ that minimize the loss 5, however the sequence fails to converge towards the true solution $p^*$ (Mandal & Apte, 2024, Proposition 4.1):

$$\lim_{n \to \infty} \left[ \frac{1}{N_\mathcal{F} \times M_\mathcal{F}} \sum_{k=1}^{N_\mathcal{F}} \sum_{j=1}^{M_\mathcal{F}} \left( \frac{\partial f_n}{\partial t}(x^k, t^j) - \mathcal{F}f_n(x^k, t^j) \right)^2 + \frac{1}{N_\mathcal{F}} \sum_{k=1}^{N_\mathcal{F}} \left( f_n(x^k, 0) - \psi(x^k) \right)^2 \right] = 0 \tag{6}$$

yet

$$\lim_{n \to \infty} f_n \neq p^*.$$

In other words, despite successfully minimizing the loss, the sequence $(f_n)_{n \in \mathbb{N}}$ does not guarantee convergence to the accurate time-dependent solution of the Fokker-Planck equation, emphasizing a significant limitation of the PINN approach. This theoretical insight serves as a basis for the development of the FlowKac which aims to address these limitations and provide a more robust framework for solving time-dependent Fokker-Planck equations.

## 4 FlowKac

In this section, we introduce FlowKac, our approach for solving the Fokker-Planck equation (FPE) by leveraging two key components: the Feynman-Kac formula and normalizing flows. Each of these elements addresses critical challenges forming a powerful framework for efficient and accurate solutions.
The Feynman-Kac formula establishes a fundamental connection between stochastic processes and partial differential equations (PDEs), enabling us to express the solutions of the FPE as expectations over stochastic paths.

Complementing this, we employ the flexible architecture of a temporal normalizing flow (Both & Kusters, 2019) that is highly effective for modeling complex probability distributions and offers the advantage of efficient sampling. This capability allows us to compute pointwise values of the FPE solution at arbitrary spatial and temporal points, eliminating the need for fixed computational grids. Consequently, FlowKac enables efficient training and accurate evaluation of the ground-truth probability density at new points of interest. By combining the Feynman-Kac formula to derive the unique solution with the expressive power of temporal normalizing flows to approximate it, FlowKac provides a robust, time-dependent, and mesh-free framework for solving the FPE. Furthermore, FlowKac addresses the limitations of numerical approximation methods (e.g., FEM, ADI) and neural network-based approaches like PINNs.

## 4.1 Temporal Normalizing flow

The temporal normalizing flow (Both & Kusters, 2019) is a flow-based model designed to approximate and model intricate time-dependent distributions. Let $X \in \mathbb{R}^d$ be a random variable and $p_X(x)$ its corresponding probability density function that we aim to model. A normalizing flow (NF) (Papamakarios et al., 2021; Kobyzev et al., 2021; Grathwohl et al., 2019) is constructed via a simple base distribution $p_Z(z)$ and a differentiable bijective mapping $f : X \to Z$ such that $X = f^{-1}(Z)$. This transformation allows for exact density estimation through the change of variable formula:

$$\log p_X(x) = \log p_Z\big(f(x)\big) + \log |\det \nabla_x f(x)| \tag{7}$$

To address the dynamics of time-dependent distributions, the temporal normalizing flow (TNF) extends this framework to the temporal domain by introducing a time-augmented variable $X^* = (X, t)$ and its corresponding latent representation $Z^* = (Z, t')$. The TNF then models the evolution of $X^*$ through a time-dependent transformation:

$$\log p_{X^*}(x^*) = \log p_{Z^*}\big(f(x^*)\big) + \log |\det \nabla_{x^*} f| \tag{8}$$

where $\nabla_{x^*} f$ denotes the Jacobian matrix of the time-augmented mapping $f(., t)$. A critical aspect to highlight is the dependence between the latent and observable variables, expressed by the relationships $z = z(x, t)$ and $t' = t'(x, t)$. Moreover, the TNF enforces the invariance of the time variable, $t' = t$. This constraint arises because the bijective transformation that maps the simple latent density into a more complex one is applied exclusively to the spatial variable $x$, ensuring that the resulting function retains the essential normalization property of a probability density. Time merely serves as an index for the transformation. As a result, the Jacobian matrix simplifies to:

$$\nabla_{x^*} f^* = \left| \begin{array}{cc} \frac{\partial \boldsymbol{z}}{\partial \boldsymbol{x}} & \frac{\partial \boldsymbol{z}}{\partial t} \\ \frac{\partial t^*}{\partial \boldsymbol{x}} & \frac{\partial t^*}{\partial t} \end{array} \right| = \left| \begin{array}{cc} \frac{\partial \boldsymbol{z}}{\partial \boldsymbol{x}} & \frac{\partial \boldsymbol{z}}{\partial t} \\ 0 & 1 \end{array} \right| = \nabla_x f$$

The TNF $f$ is constructed through a sequential chaining of multiple transformations:

$$z^* = f(x, t) = T_{[L]} \circ T_{[L-1]} \circ ... \circ T_{[1]}(x, t) \quad \text{and} \quad x^* = f^{-1}(z, t) = T_{[1]}^{-1} \circ T_{[2]}^{-1} \circ ... \circ T_{[L]}^{-1}(z, t).$$

Each of these transformations adheres to the KRnet architecture (Tang et al., 2021) which is detailed in Appendix A. The architecture comprises two fundamental layers: an Actnorm layer that scales the input and adds a bias, followed by an affine coupling layer adapted from real NVP (Dinh et al., 2017) and responsible for mixing the spatial and temporal variables. KRnet adds a nonlinear transformation at the end of this sequence, helping to capture more complex and challenging dynamics, which constitutes an improvement over RealNVP. KRnet also offers key advantages over ODE-based normalizing flows, such as Continuous Normalizing Flows (CNFs) (Chen et al., 2018). In CNFs, transformations are governed by neural ODEs, where the mixing of spatial and temporal variables occurs implicitly during ODE integration. In contrast, KRnet explicitly controls the mixing of spatial and temporal variables through its discrete, structured transformations, allowing for faster training while maintaining flexibility in modeling complex probability distributions. The Jacobian matrix of this sequential flow is derived via the chain rule:

$$|\det \nabla_x f| = \prod_{i=1}^{L} \left| \det \nabla_{x_{[i-1]}} T_{[i]} \right|$$

here $x_{[i]}$ is the output of each transformation, with $x_{[0]} = x$ and $x_{[L]} = z$.

## 4.2 Training algorithm

Our purpose is to train the TNF to solve the time-dependent Fokker-Planck equation accurately. As discussed in Section 3.3, the Physics-Informed approach can suffer from convergence issues, thus simply minimizing the PINN loss given by Equation 5 is insufficient to guarantee an accurate solution. Instead, following work in Beck et al. (2021b) we leverage the Feynman-Kac formula (Oksendal, 2013, Theorem 8.2.1) to guide the training process:

**Theorem 4.1 (The Feynman-Kac formula)** *Let $(\tilde{X}_t)$ be an Ito process with drift $\tilde{\mu}$ and diffusion $\tilde{\sigma}$. Let $q \in C(\mathbb{R}^d)$ and $v \in C^{2,1}(\mathbb{R}^d \times \mathbb{R}^+)$ satisfy for all $(x,t) \in \mathbb{R}^d \times [0,T]$:*

$$-\frac{\partial v}{\partial t} + \sum_{i=1}^d \tilde{\mu}_i \frac{\partial v}{\partial x_i} + \frac{1}{2} \sum_{i,j=1}^d \left( \tilde{\sigma} \tilde{\sigma}^T \right)_{ij} \frac{\partial^2 v}{\partial x_i x_j} = qv \tag{9}$$

*with initial condition $v(x,0) = f(x)$ such that $f \in C_0^2(\mathbb{R}^d)$. Then, the unique solution can be expressed as:*

$$v(x,t) = \mathbb{E}\left[ \exp\left( -\int_0^t q(\tilde{X}_s, s) ds \right) f\left( \tilde{X}_t \right) \Big| \tilde{X}_0 = x \right].$$

Such a formula provides a powerful connection between parabolic partial differential equations (such as the FPE) and expectations of stochastic processes.

This representation allows us to define a more effective loss criterion for the training process based on pathwise expectations. To apply the Feynman-Kac formula, we first rewrite the Fokker-Planck equation 2 into a form matching equation 9, isolating a term that corresponds to the function $q$ (see Appendix B for derivation and more details):

$$-\frac{\partial p}{\partial t} + \sum_{i=1}^d \left( -\mu_i + 2 \sum_{j=1}^d \frac{\partial D_{ij}}{\partial x_j} \right) \frac{\partial p}{\partial x_i} + \sum_{i,j=1}^d D_{ij} \frac{\partial^2 p}{\partial x_i \partial x_j} = \left( \nabla \cdot \mu - \nabla \cdot (\nabla \cdot D) \right) p \tag{10}$$

We identify the scalar function $q = \nabla \cdot \mu - \nabla \cdot (\nabla \cdot D)$ as the effective potential term in the transformed PDE. Then, we apply the Feynman-Kac formula to the transformed equation 10 to obtain a solution of the form:

$$p_{\text{FK}}(x,t) = \mathbb{E}\left[ \exp\left( -\int_0^t q\left( \tilde{X}_s, s \right) ds \right) \psi\left( \tilde{X}_t \right) \mid \tilde{X}_0 = x \right] \tag{11}$$

here $(\tilde{X}_t)_{t \geq 0}$, referred to as the FlowKac process, represents a stochastic process characterized by the drift and diffusion terms:

$$\begin{cases} \tilde{\mu} = \left[ -\mu_i + 2 \sum_{j=1}^d \frac{\partial D_{ij}}{\partial x_j} \right]_i^\top \\ \tilde{\sigma} = \sigma. \end{cases} \tag{12}$$

Finally, to optimize the parameters $\theta$ of the TNF, we propose the following loss function:

$$\mathcal{L}(\theta) = \int_0^T \|p_\theta(.,t) - p_{\text{FK}}(.,t)\|_2^2 dt = \int_0^T \int_{\mathbb{R}^d} \left( p_\theta(x,t) - p_{\text{FK}}(x,t) \right)^2 dx \, dt \tag{13}$$

This regression loss function measures the discrepancy between the TNF's predicted density $p_\theta$ and the true solution derived via the Feynman-Kac formula $p_{\text{FK}}$, with the $L^2$ norm inducing a strong form of convergence

which also implies the convergence in distribution of the underlying stochastic process. Thus, the TNF is guided toward the correct solution by minimizing this loss.

However, directly discretizing the integral given by Equation 13 is computationally prohibitive for high-dimensional systems, as it requires covering a $d$-dimensional mesh, causing the computational complexity to scale exponentially.

To alleviate this constraint, we use a sampling-based approximation that bypasses the mesh-based evaluation. Specifically, we uniformly sample $n_x$ points from a spatial domain $[a, b]^d$ sufficiently large to encompass the support of the distribution for all $t$, and $n_t$ temporal points from $[0, T]$. This reduces the computational cost by replacing the uniform discretization of the integral with a discrete sum over sampled points:

$$\hat{\mathcal{L}}(\theta) = \frac{1}{n_x n_t} \sum_{k=1}^{n_x} \sum_{j=1}^{n_t} \left( p_\theta(x^k, t^j) - p_{\text{FK}}(x^k, t^j) \right)^2 \tag{14}$$

We further enhance this approach by leveraging the generative properties of normalizing flows. Instead of relying solely on uniform sampling, we use an adaptive sampling mechanism that incorporates learned density information (Gabrié et al., 2022). Specifically, after an initial training phase, we refine the training set by inverting the normalizing flow, thereby generating $n_x$ new spatial samples that better reflect the underlying probability density. This adaptivity concentrates computational effort on regions of higher probability mass, improving both training efficiency and density estimation accuracy. Quantitative comparisons between uniform and adaptive sampling are provided in Appendix G.

A critical advantage of this approach lies in the properties of the Feynman-Kac formula, which enables the computation of pointwise solutions to the Fokker-Planck equation without relying on adjacent mesh points. By eliminating mesh dependencies, this sampling strategy significantly mitigates the curse of dimensionality while maintaining high fidelity in the approximation of the loss function.

The proposed training procedure is detailed in Algorithm 1: we sample multiple stochastic paths starting from each training point, compute the Feynman-Kac-based density estimate, and evaluate the corresponding loss. The process is repeated across all training samples.

---

**Algorithm 1:** FlowKac (naive)

**Input** : Maximum epochs $N_e$, number of sample paths $n_W$, number of spatial points $n_x$, number of temporal points $n_t$

1 **for** $l = 1, \ldots, N_e$ **do**
2     Sample uniformly $C_{train} = (x^k, t^j)_{1 \leq k \leq n_x, 1 \leq j \leq n_t}$;
3     Initialize loss $L_e = 0$;
4     **for** $k = 1, \ldots, n_x$ **do**
5         Sample $n_W$ paths of $\tilde{X}_t$ at time points $(t^j)_{1 \leq j \leq n_t}$ starting from $x^k$;
6         Compute $p_{\text{FK}}(x^k, t^j)$ using empirical mean;
7         Compute TNF's output $p_\theta(x^k, t^j)$;
8         Accumulate loss: $L_e = L_e + \sum_j \left( p_\theta(x^k, t^j) - p_{\text{FK}}(x^k, t^j) \right)^2$;
9     **end for**
10     Update model parameters $\theta$ using the Adam optimizer;
11 **end for**

**Output:** The predicted solution $p_\theta(x, t)$

---

The major computational challenge in the training process lies in the repeated application of the Feynman-Kac formula for each training sample. Since sampling the stochastic process $(\tilde{X}_t)$ is conditioned on the initial point $x^k$, a new set of stochastic paths must be generated for every individual data point, increasing computational complexity. This challenge is inherent to all methods based on Feynman-Kac reformulation and path sampling, where the necessity of repeatedly simulating stochastic trajectories leads to an increase

in computational cost, particularly in high-dimensional settings. As a result, this bottleneck highlights the critical need for more computationally efficient approaches.

In the following section, we introduce an enhanced version of the algorithm that significantly accelerates the training process by addressing this inefficiency.

### 4.3 Stochastic sampling trick

In this section, we introduce a novel and efficient algorithm for sampling the FlowKac process $\tilde{X}$ by leveraging the properties of stochastic flows (Kunita & Kunita, 1990). The SDE governing $(\tilde{X}_t)_{t\in[0,T]}$ is given by:

$$d\tilde{X}_t = \tilde{\mu}(\tilde{X}_t, t)dt + \sigma(\tilde{X}_t, t)dW_t, \tag{15}$$

We denote by $\Phi_{s,t}(x) = \tilde{X}_t^{s,x}$ the solution to the SDE 15 starting from $x$ at time $s$. $\Phi_{s,t}$ defines a stochastic flow.
An important property of stochastic flows is stated by the following theorem (Protter, 2005, Theorem 40):

**Theorem 4.2** *Let drift $\tilde{\mu}$ and diffusion $\sigma$ coefficients have locally Lipschitz derivatives up to order $k$. Then, for any fixed realization $\omega \in \Omega$, the mapping $\Phi_{s,t}(.,\omega) : \mathbb{R}^d \to \mathbb{R}^d$ is $k$ times continuously differentiable.*

This theorem establishes that the solution to Equation 15 is smooth with respect to the initial condition, thus allowing us to employ high-order Taylor expansions (Cartan, 1967, Theorem 5.6.1 and 5.6.3) to the stochastic flow and expressing solutions to the SDE starting from arbitrary initial points $x = x_0 + h$, leveraging a solution starting from a reference point $x_{\text{ref}}$:

$$\Phi_{0,t}(x) = \Phi_{0,t}(x_{\text{ref}}) + \Phi'_{0,t}(x_{\text{ref}}).(h) + \frac{1}{2}\Phi''_{0,t}(x_{\text{ref}}).(h,h) + ... + \frac{1}{(k)!}\Phi^{(k)}_{0,t}(x_{\text{ref}}).(h)^k + o\left(\|h\|^k\right). \tag{16}$$

here, $\Phi'_{0,t}(x_{\text{ref}}).(h) = J_{\Phi,t}(x_{\text{ref}})h$ represents the linear term involving the Jacobian matrix $J_{\Phi,t}$, while $\Phi''_{0,t}(x_{\text{ref}}).(h,h) = h^\top H_{\Phi,t}(x_{\text{ref}})h$ denotes the quadratic term, involving the Hessian tensor $H_{\Phi,t}$. Higher-order terms provide additional precision for approximating the stochastic flow but are typically truncated for computational efficiency.

For dynamics that depend linearly on the initial condition, the stochastic sampling trick provides exact solutions, regardless of the choice of $x$ and $x_{\text{ref}}$, as illustrated in the 1-dimensional geometric Brownian motion (GBM) example in Section 5.1. In these cases, all higher-order terms in the Taylor expansion vanish since they are independent of the initial condition $x_{\text{ref}}$, resulting in an exact reconstruction of the solution.

This expansion forms the cornerstone of our novel sampling algorithm as we can efficiently generate samples of $\tilde{X}_t$ starting from perturbed initial conditions while maintaining high accuracy. This approach significantly reduces the computational burden of sampling the FlowKac SDE for each data point as summarized in Algorithm 2.

However, for dynamics with nonlinear dependencies with respect to the initial condition, maintaining accuracy requires careful selection of the reference point $x_{\text{ref}}$ in the Taylor expansion. To preserve the local validity of the expansion and avoid significant approximation errors, we propose a variant where $x_{\text{ref}}$ is dynamically chosen as the centroid of each mini-batch. The resulting modified training algorithm is detailed in Appendix K, as shown in Algorithm 4.

## 5   Numerical results

To assess the performance of our proposed model, we conducted a series of experiments across a range of stochastic differential equations. We begin with a simple 1-dimensional example to validate the foundational aspects of our approach. Following this, we apply our method to more complex processes then to nonlinear SDEs, demonstrating its robustness in capturing intricate dynamics. Finally, we extended the framework

---

**Algorithm 2:** FlowKac (stochastic sampling trick)

---

**Input** : Maximum epochs $N_e$, number of sample paths $n_W$, number of spatial points $n_x$, number of temporal points $n_t$, reference point for Taylor expansion $x_{\text{ref}}$

**1 for** $l = 1, \ldots, N_e$ **do**

**2**  Sample uniformly $C_{train} = (x^k, t^j)_{1 \leq k \leq n_x, 1 \leq j \leq n_t}$;

**3**  Sample $n_W$ Brownian motion paths $W_t^l$ ;  `// Fix realization` $\omega$ `across all training points as required by Theorem 4.2`

**4**  Compute Jacobian $J_{\Phi,t}(x_{\text{ref}})$ and Hessian $H_{\Phi,t}(x_{\text{ref}})$ using automatic differentiation;

**5**  Divide $C_{\text{train}}$ into $m$ batches $\{C^b\}_{b=1}^m$;

**6**  Initialize loss $L_e = 0$;

**7**  **for** $b = 1, \ldots, m$ **do**

**8**    Compute Taylor expansion for batch $C^b$ starting from $x_{\text{ref}}$ (Equation 16);

**9**    Compute $p_{\text{FK}}$;

**10**    Compute batch loss $L_b$;

**11**    Accumulate loss: $L_e = L_e + L_b$;

**12**  **end for**

**13**  Update model parameters $\theta$ using the Adam optimizer;

**14 end for**

**Output:** The predicted solution $p_\theta(x, t)$

---

to tackle higher-dimensional stochastic processes, showcasing our model's scalability and effectiveness in handling increasingly complex systems. Our model is compared to the PINN-TNF approach of Feng & Zhou (2022). The comparison is based on both qualitative and quantitative assessments. Qualitatively, we give, for each SDE, probability density and marginal distribution plots to visualize the fidelity of the learned solutions. Quantitatively, following Feng & Zhou (2022) and Tang et al. (2021), we compute metrics based on the relative $L^2$ distance and the Kullback-Leibler (KL) divergence between the ground truth probability density $p^*$ and the model's predicted density $p_\theta$. All quantitative results are then summarized in a dedicated discussion subsection 5.6:

$$L^2(t) = \frac{\|p^*(.,t) - p_\theta(.,t)\|_2^2}{\|p^*(.,t)\|_2^2}, \quad D_{\text{KL}}\big(p^*(.,t)\|p_\theta(.,t)\big) = \int p^*(x,t) \log \frac{p^*(x,t)}{p_\theta(x,t)} dx.$$

To ensure accurate computation, both metrics are evaluated on a spatial grid, leveraging $n_{eval}$ evaluation points uniformly spaced across the support of $p^*(.,t)$ ensuring thorough coverage of the probability density for SDEs up to four dimensions. For higher-dimensional examples, where a uniform grid becomes computationally infeasible, we instead draw a fixed number of sample points directly from the ground-truth density to compute the metrics efficiently:

$$\tilde{L}^2(t) = \frac{\sum_{k=1}^{n_{eval}} \big(p^*(x^k,t) - p_\theta(x^k,t)\big)^2}{\sum_{k=1}^{n_{eval}} p^*(x^k,t)^2}, \quad \tilde{D}_{\text{KL}}(t) = \frac{1}{n_{eval}} \sum_{k=1}^{n_{eval}} p^*(x^k,t) \log \frac{p^*(x^k,t)}{p_\theta(x^k,t)}.$$

For all numerical experiments, we employ the Adam optimizer (Kingma & Ba, 2014) with a learning rate of $\eta = 0.001$ and default momentum parameters to optimize the normalizing flow parameters. The stochastic differential equations are numerically sampled using the `torchsde` framework of Kidger et al. (2021); Li et al. (2020), which implements adaptive-step solvers based on the Euler-Maruyama and higher-order strong schemes. Finally, for the stochastic sampling trick, the Jacobian and Hessian terms are computed efficiently using the automatic differentiation capabilities of `pytorch` (see details in Appendix E).

We also include in Appendix J a dedicated comparison between FlowKac and an MLP trained directly on the Feynman–Kac loss, highlighting the benefits of incorporating a generative model and isolating them from those of the Feynman–Kac formulation.

### 5.1 Univariate Geometric Brownian Motion

We first apply our framework to the univariate Geometric Brownian Motion (GBM). This process serves as an ideal test case due to the existence of a known analytical closed-form solution, providing a rigorous benchmark for our methodology. The following SDE characterizes the GBM dynamics:

$$dX_t = \mu X_t dt + \sigma X_t dW_t,$$

where $\mu$ represents the constant drift coefficient, $\sigma$ denotes the constant volatility, and $W_t$ is a standard one-dimensional Wiener process. The corresponding FPE is described as follows:

$$\frac{\partial p}{\partial t} = (-\mu + \sigma^2)p + (-\mu x + 2\sigma^2 x)\frac{\partial p}{\partial x} + \frac{1}{2}\sigma^2 x^2 \frac{\partial^2 p}{\partial x^2}. \tag{17}$$

We choose the following initial condition:

$$\psi(x) = \frac{1}{x\sqrt{2\pi\sigma^2}} \exp\left(-\frac{\log(x)^2}{2\sigma^2}\right).$$

The exact solution for the process is given by:

$$X_t = X_0 \exp\left(\left(\mu - \frac{1}{2}\sigma^2\right)t + \sigma W_t\right),$$

with the associated probability density function:

$$p^*(x,t) = \frac{1}{x\sqrt{2\pi(t+1)\sigma^2}} \exp\left(-\frac{\left(\log(x) - \left(\mu - \frac{1}{2}\sigma^2\right)t\right)^2}{2(t+1)\sigma^2}\right) \tag{18}$$

We solve the FPE on the interval $(0,5]$ using a training set of size $|C_{\text{train}}| = 4.10^4$ generated through a uniform distribution, and $n_w = 500$ sample paths. The model is trained for $N_e = 200$ epochs, and for our normalizing flow, we use a sequence of depth $L = 8$. Notably, we employ our stochastic sampling trick by computing a first-order Taylor expansion to efficiently sample the resulting FlowKac process (Equation 12):

$$d\tilde{X}_t = (-\mu + 2\sigma^2)\tilde{X}_t dt + \sigma\tilde{X}_t dW_t, \tag{19}$$

To compute $\tilde{\Phi}_{0,t}(x,\omega)$, the stochastic flow solution to Equation 19 originating from $x$ at $t = 0$, we first derive the Jacobian with respect to the reference point (see details in Appendix E):

$$J_{\tilde{\phi},t}(x_{\text{ref}}) = \exp\left(\left(-\mu + \frac{3\sigma^2}{2}\right)t + \sigma W_t\right)$$

In this example, a first-order Taylor expansion turns out to be sufficient to completely capture the dynamics, as all higher-order terms vanish due to their independence from the reference point, and yields:

$$\tilde{X}_t^x = \tilde{\Phi}_{0,t}(x) = \tilde{\Phi}_{0,t}(x_{\text{ref}}) + (x - x_{\text{ref}}).J_{\phi,t}(x_{\text{ref}}) \tag{20}$$

$$= x \exp\left(\left(-\mu + \frac{3\sigma^2}{2}\right)t + \sigma W_t\right).$$

The stochastic sampling trick represents a significant speed-up in our computational approach, allowing us to compute sample paths of Equation 19 with perfect accuracy while significantly reducing the computational cost. Table 1 presents a comprehensive performance comparison between the standard FlowKac implementation and the optimized version incorporating the stochastic sampling technique. The results reveal consistent and substantial speedups across different training set sizes and sampling configurations. Additional runtime comparisons for other SDEs are provided in Appendix I.

In practical implementations, our experiments were conducted using 300 to 500 stochastic paths per training point, which proved sufficient for producing accurate density estimations. Notably, as the dimensionality

| Configuration | | Run Time (seconds) | | Speedup |
|---|---|---|---|---|
| Training Size $(|C_{\text{train}}|)$ | $W_t$ Samples $(n_W)$ | FlowKac (naive) | FlowKac (Sampling trick) | Factor |
| $2 \times 10^4$ | 500 | 31.1 | **3.9** | 8× |
| $2 \times 10^4$ | 1,000 | 57.7 | **5.2** | 11× |
| $6 \times 10^4$ | 500 | 93.1 | **6.6** | 14× |
| $6 \times 10^4$ | 1000 | 171.5 | **10.5** | 16× |

Table 1: Performance comparison between naive FlowKac and FlowKac with stochastic sampling trick, for a single training epoch using a fixed batch size of 2000. The experiments were conducted on an NVIDIA T4 GPU with 16 GB memory. The speedup factor demonstrates increasing efficiency gains with larger training sets, highlighting the technique's scalability benefits.

of the SDE increases, the required training size will also grow accordingly to maintain accuracy. In this context, the benefits of the stochastic sampling trick become even more pronounced, mitigating the otherwise prohibitive computational burden associated with high-dimensional sampling.

The results, displayed in Figure 2, show the comparison between the predicted density and the exact solution from Equation 18. This comparison demonstrates strong alignments between the predicted density and the exact solution, validating the accuracy of our framework for this univariate case.

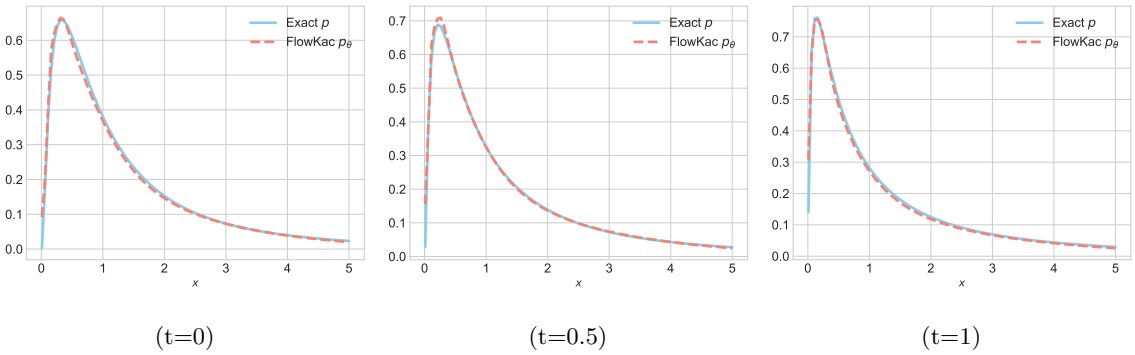

(t=0)  (t=0.5)  (t=1)

Figure 2: Comparison between FlowKac density $p_\theta$ and the true density of the univariate GBM process, at $t = 0$, $t = 0.5$ and $t = 1$.

## 5.2 Multivariate Ornstein-Uhlenbeck

We next explore the dynamics of a two-dimensional Ornstein-Uhlenbeck (OU) process governed by The following SDE:

$$dX_t = AX_t dt + \Sigma dW_t \tag{21}$$

$$= \begin{pmatrix} 0.1 & 1 \\ -1 & -0.1 \end{pmatrix} X_t dt + \begin{pmatrix} 0.6 & 0 \\ 0 & 0 \end{pmatrix} dW_t. \tag{22}$$

and the initial condition is given as:

$$\psi = \mathcal{N}\left(\begin{pmatrix} 1 \\ 1 \end{pmatrix}, \frac{1}{9}I_2\right) \tag{23}$$

The unique strong solution to Equation 21 can be expressed as follows (Gobet & She, 2016, Proposition 1):

$$X_t = e^{At} \left[ X_0 + \int_0^t e^{-As} \Sigma dW_s \right] \tag{24}$$

its mean $m_t$ and covariance matrix $V_t$ are given by:

$$m_t := \mathbb{E}[X_t] = e^{At} \mathbb{E}[X_0] \tag{25}$$

$$V_t := \mathbb{E}[X_t X_t^\top] = e^{At} \left( \mathbb{E}[X_0 X_0^\top] + \int_0^t e^{-As} \Sigma \Sigma^\top e^{-A^\top s} \, ds \right) e^{A^\top t} \tag{26}$$

Therefore, the process $(X_t)$ follows a multivariate normal distribution, and the exact solution of the corresponding FPE is:

$$p^*(.,t) = \mathcal{N}(m_t, V_t).$$

We solve the FPE on the interval $[-5,5]^2$ using a training set of size $|C_{\text{train}}| = 4.10^4$ generated through the uniform distribution. The model is trained for $N_e = 250$ epochs. For the normalizing flow, we use a sequence of depth $L = 8$.

The results presented in Figure 3 show the comparison of the predicted 2-dimensional density at different time points ($t = 0, 1, 2,$ and $3$) for the FlowKac model, the PINN model, and the exact solution derived from Equation 24. The FlowKac model demonstrates a strong capability to accurately replicate, across time, the behavior of the process. In contrast, the PINN model shows a significant deviation from the true density, particularly at later time points.

Quantitative results, as summarized in Table 2 and further emphasized in Table 3 highlight a significant performance gap between the two approaches. The PINN model demonstrates a progressive deterioration in accuracy as proven by the increasing error metrics over time ($L^2$ error increasing from $5.7 \times 10^{-3}$ to $1.11$) reflecting a divergent behavior. In contrast, FlowKac maintains stable and lower error values throughout the simulation.

This reinforces the robustness and precision of our framework in modeling complex stochastic processes.

## 5.3 Multivariate Geometric Brownian Motion

The third process is a two-dimensional extension of the Geometric Brownian motion (Barrera et al., 2022), and is described by the following SDE:

$$dX_t = \left( A + \frac{1}{2} B^2 \right) X_t dt + B X_t dW_t \tag{27}$$

where $A, B \in \mathbb{R}^{2 \times 2}$ such that the eigenvalues of $(A + \frac{1}{2} B^2)$ have a negative real part, and $W_t$ is a univariate Brownian motion. Under these conditions, Equation 27 admits a closed-form solution:

$$X_t = \exp\left( tA + BW_t \right) X_0 \tag{28}$$

here $A = \begin{pmatrix} a_1 & 0 \\ 0 & a_2 \end{pmatrix} = \begin{pmatrix} -1 & 0 \\ 0 & -2 \end{pmatrix}$, $B = \begin{pmatrix} b_1 & 0 \\ 0 & b_2 \end{pmatrix} = \begin{pmatrix} 0.5 & 0 \\ 0 & 1 \end{pmatrix}$ and the initial state $X_0$ is distributed according to the multivariate log-normal distribution:

$$\psi = \text{Log-}\mathcal{N} \left( \mu_0 = \begin{pmatrix} 0.5 \\ 0.7 \end{pmatrix}, \Sigma_0 = \frac{1}{2} I_2 \right)$$

The exact solution in this instance can be expressed in a closed-form formula (details in Appendix D):

$$p^*(.,t) = \text{Log-}\mathcal{N} \left( \mu_t = \begin{pmatrix} 0.5 + a_1 t \\ 0.7 + a_2 t \end{pmatrix}, \Sigma_t = \begin{pmatrix} 0.5 + (b_1)^2 t & b_1 b_2 t \\ b_1 b_2 t & 0.5 + (b_2)^2 t \end{pmatrix} \right), \tag{29}$$

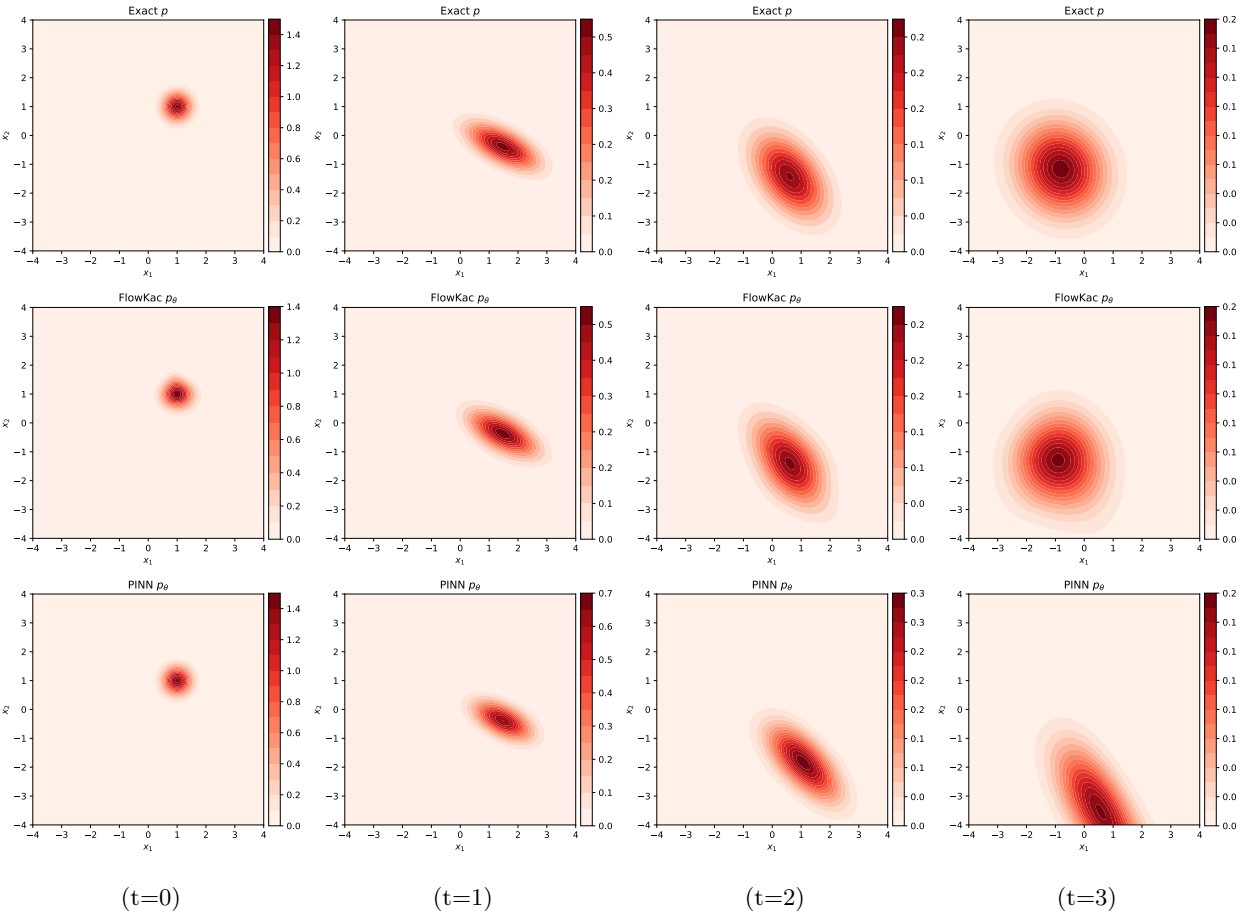

Figure 3: Comparison of density distributions for the 2D Ornstein-Uhlenbeck process at $t = 0$, $t = 1$, $t = 2$, and $t = 3$. The exact solution is depicted in the top row, FlowKac predictions in the middle row, and PINN predictions in the bottom row.

this density converges to a stationary Dirac distribution centered at 0 as $t \to \infty$.

To solve the associated FPE on the interval $(0, 6]^2$, we employ a training set of size $|C_{\text{train}}| = 6.10^4$ generated through the uniform distribution. The model is trained over $N_e = 300$ epochs, using a normalizing flow of depth $L = 14$. An ablation study analyzing the impact of varying the flow's depth, layer width, and learning rate on performance is presented in Appendix H.

Results depicting the two-dimensional densities are displayed in Figure 4. The comparison shows a strong alignment between the true solution of the Fokker-Planck equation and the prediction generated by FlowKac. As the density evolves towards the stationary state consisting of a Dirac distribution centered at 0, the support of the distribution becomes progressively concentrated. Such a configuration presents significant challenges for mesh-based approximation methods, which struggle to capture such a sharply localized distribution. Additionally, alternative Feynman-Kac reformulations relying on the steady-state solution would be ineffective for this SDE due to the singular behavior of the Dirac distribution.

Quantitative results in Table 2 reinforce the qualitative observations. Specifically, FlowKac demonstrates lower $L^2$ errors and $D_{\text{KL}}$ values when compared to the PINN approach, further validating our model.

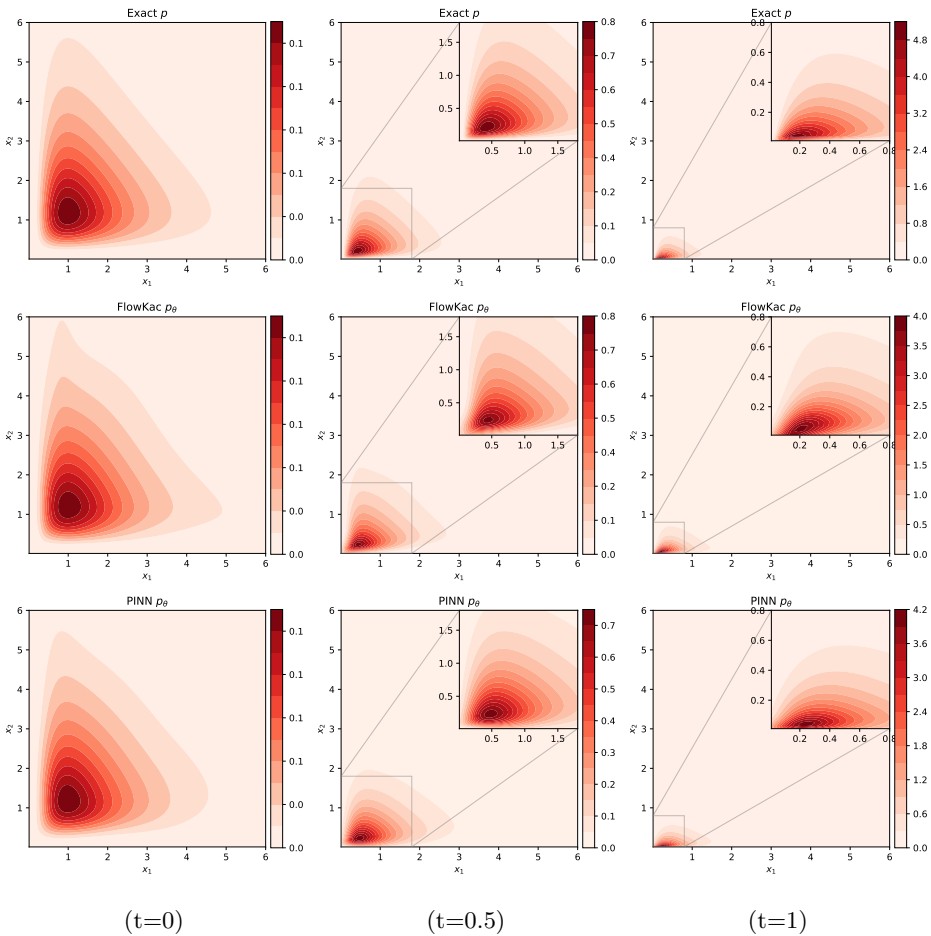

Figure 4: Comparison of 2D density distributions for the Multivariate GBM at $t = 0$, $t = 0.5$, and $t = 1$. The exact solution is depicted in the top row, FlowKac predictions in the middle row, and PINN predictions in the bottom row.

## 5.4 Duffing oscillator

To further assess the ability of our model to capture complex chaotic nonlinear dynamics, we solve the FPE for the 2-dimensional Duffing oscillator (Pichler et al., 2013) described by:

$$\begin{pmatrix} dX_{1t} \\ dX_{2t} \end{pmatrix} = \begin{pmatrix} X_{2t} \\ -0.4\omega X_{2t} + \omega^2 X_{1t} - 0.1\omega^2 X_{1t}^3 \end{pmatrix} dt + \begin{pmatrix} 0 & 0 \\ 0 & \sqrt{0.8} \end{pmatrix} dW_t. \tag{30}$$

with initial condition $\psi = \mathcal{N}\left(\begin{pmatrix} 0 \\ 8 \end{pmatrix}, \frac{1}{2} I_2\right)$.

Since no closed-form solution exists for this system, the ground truth is approximated numerically by solving the FPE using the ADI scheme, following the approach in Feng & Zhou (2022). To ensure robustness and accuracy, the computation is performed on a finely discretized mesh over the domain $[-10, 10]^2$ (implementation details provided in Appendix C). This high-resolution numerical solution serves as a reliable reference, enabling rigorous evaluation of FlowKac model's output.

For FlowKac, we train the model to solve the FPE associated with Equation 30, for $\omega = 1$, on the same interval $[-10, 10]^2$. The training dataset consists of $|C_{\text{train}}| = 3 \times 10^4$ sample drawn uniformly across the domain. The model is trained over $N_e = 300$ epochs, using a normalizing flow architecture of depth $L = 10$ and by employing the nonlinear layer with 60 bins (details in Appendix A). Notably, for this example, we simulate the FlowKac SDE directly rather than utilizing the stochastic sampling trick. This decision

stems from the need for high-order derivatives (higher than second-order) in the Taylor expansion for the sampling trick to work. These high-order derivatives are computationally intensive to estimate using sampling approximations.

Figure 5 compares the solutions obtained with FlowKac, PINNs, and the ground truth computed using the ADI scheme. We again observe a strong alignment between the predicted densities by FlowKac and the exact solutions. Additionally, the quantitative results in Table 2 further highlight FlowKac's superior accuracy over the PINN approach, as demonstrated by consistently lower $L^2$ and $D_{\mathrm{KL}}$ values across time.

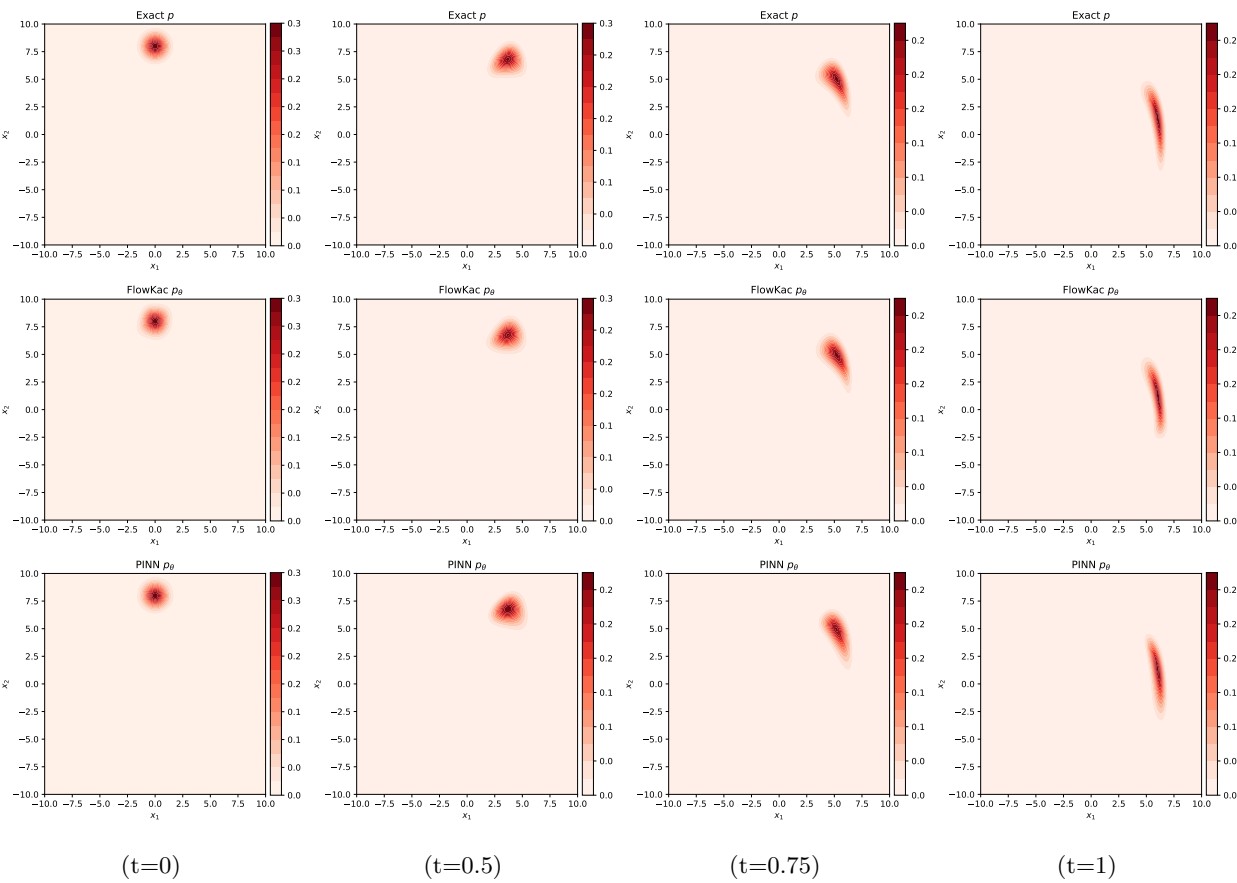

Figure 5: Comparison of 2D density distributions for the Duffing non-linear oscillator at $t = 0$, $t = 0.5$, $t = 0.75$, and $t = 1$. The exact solution, given by ADI, is depicted in the top row, FlowKac predictions in the middle row, and PINN predictions in the bottom row.

## 5.5 Higher dimensional examples

Finally, to assess the scalability of our approach, we consider a relatively higher-dimensional Ornstein-Uhlenbeck process:

$$dX_t = AX_t dt + \Sigma dW_t \tag{31}$$

where $A = aI_d$, $\Sigma = \sigma I_d$ and $W_t$ is a $d$-dimensional Brownian motion. The model's structure allows for closed-form solutions and exact marginal distributions in all dimensions, making it particularly well-suited for high-dimensional analysis. The associated FPE is given by:

$$\frac{\partial p}{\partial t} = -a \sum_{i=1}^{d} (x_i \frac{\partial p}{\partial x_i} + p) + \frac{1}{2}\sigma^2 \sum_{i,j=1}^{d} \frac{\partial^2 p}{\partial x_i \partial x_j}. \tag{32}$$

For the initial condition, we assume a multivariate normal distribution:

$$\psi = \mathcal{N}\left(m_0 = \begin{bmatrix} 1 \\ \vdots \\ 1 \end{bmatrix}, V_0 = \frac{1}{4}I_d\right)$$

The exact solution is given by:

$$p(.,t) = \mathcal{N}\left(m_t = e^{at}m_0, V_t = e^{2at}V_0 + \frac{\sigma^2}{2a}(e^{2at} - 1)I_d\right) \tag{33}$$

Figure 6 presents a comparative analysis between FlowKac and PINN models for the 4-dimensional setting, demonstrating the superior performance of our model, which consistently achieves lower $D_{\mathrm{KL}}$ and $L^2$ errors across various time values.

As dimensionality increases, computing the PINN loss becomes increasingly tedious. Consequently, we report performance metrics exclusively for FlowKac up to $d = 12$, while noting that our model runs out-of-memory on an A100 GPU with 80 GB of memory at $d \approx 100$. To further illustrate the accuracy of FlowKac in higher dimensions, Figure 7 compares the learned 2D marginal distribution - the marginal dimensions are chosen randomly - with the ground truth at various time steps for the 8-dimensional setting of the OU process. The learned marginal distribution is obtained by sampling points from the temporal normalizing flow, and the results indicate strong agreement with the exact distribution, reinforcing the scalability of our approach and its ability to handle high-dimensional stochastic processes efficiently. Additional evaluation metrics across various dimensions are provided in Appendix F.

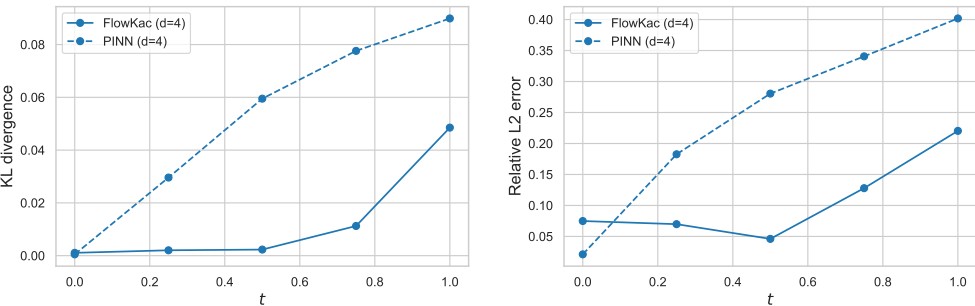

Figure 6: Comparison of $D_{\mathrm{KL}}$ and relative $L^2$ errors between FlowKac and PINN models for a 4-dimensional Ornstein-Uhlenbeck at various time values.

## 5.6 Discussion

This section provides a quantitative evaluation of FlowKac's performance compared to the PINN framework, using KL-divergence ($D_{\mathrm{KL}}$) and relative $L^2$ errors as metrics. While Table 2 offers an overview across different SDEs and time points in $t \in [0, 1]$, Table 3 specifically focuses on the 2D Ornstein-Uhlenbeck process over an extended time horizon to assess long-term stability. Our analysis reveals several key findings:

- For low-dimensional processes (e.g., 1D GBM), both approaches achieve strong performance, with PINN showing marginally better accuracy ($D_{\mathrm{KL}} = 1.30 \times 10^{-3}$ vs $2.60 \times 10^{-3}$). However, the practical significance of this difference is minimal given that the error levels are well within acceptable bounds.

- In higher-dimensional settings (2D GBM and OU), FlowKac maintains consistent convergence properties and stable error metrics, whereas PINNs exhibit growing instability over time, particularly in the 2D OU process, where errors increase exponentially (from $D_{\mathrm{KL}}(0) = 10^{-4}$ to $D_{\mathrm{KL}}(3) = 1.68$).

- FlowKac consistently preserves error values within the $10^{-2}$ to $10^{-3}$ range across all time points, demonstrating superior stability compared to PINNs.

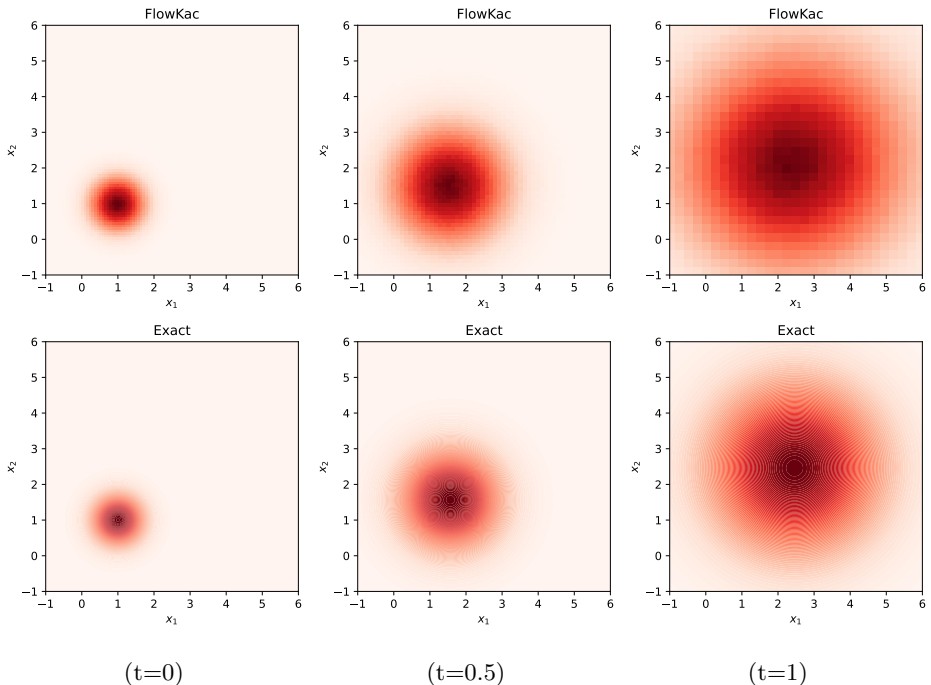

Figure 7: Comparison of 2D marginal distributions between FlowKac and the ground truth at $t = 0$, $t = 0.5$ and $t = 1$.

The PINN's observed behavior aligns with the divergence phenomenon as presented by Mandal & Apte (2024). This limitation manifests even when the training loss given by Equation 5 appears to decrease significantly. In contrast, FlowKac's consistent performance across all test cases demonstrates its robustness as a general-purpose solver for stochastic processes, outperforming PINNs in challenging scenarios.

Finally, it is worth noting that although the Feynman–Kac representation permits direct pointwise estimation of the solution, such evaluations require simulating many stochastic paths. Once trained, FlowKac provides density evaluations and samples at negligible marginal cost, thereby amortizing the initial training cost over a large number of queries. A brief runtime comparison in Appendix L quantifies this efficiency advantage.

| SDE | Model | $D_{\mathrm{KL}}(t)$ | | | | | $\mathbf{L}^2(t)$ | | | | |
|---|---|---|---|---|---|---|---|---|---|---|---|
| | | $t = 0.0$ | $t = 0.25$ | $t = 0.5$ | $t = 0.75$ | $t = 1.0$ | $t = 0.0$ | $t = 0.25$ | $t = 0.5$ | $t = 0.75$ | $t = 1.0$ |
| GBM 1d | FlowKac | 2.60e-3 | 1.90e-3 | 1.80e-3 | 2.00e-3 | 2.60e-3 | 6.28e-2 | 3.56e-2 | 3.82e-2 | 4.61e-2 | 5.93e-2 |
| | PINN | **3.00e-4** | **9.00e-4** | **1.50e-3** | **1.60e-3** | **1.30e-3** | **5.20e-3** | **1.94e-2** | **2.53e-2** | **3.08e-2** | **3.44e-2** |
| Ornstein-Uhlenbeck 2d | FlowKac | 1.60e-2 | **1.36e-2** | **1.22e-2** | **1.13e-2** | **8.90e-3** | 6.07e-2 | **7.79e-2** | **5.73e-2** | **4.88e-2** | **4.70e-2** |
| | PINN | **1.00e-4** | 1.65e-2 | 3.25e-2 | 4.37e-2 | 4.94e-2 | **5.70e-3** | 1.13e-1 | 1.58e-1 | 1.89e-1 | 2.10e-1 |
| GBM 2d | FlowKac | 1.98e-2 | **1.30e-2** | **1.92e-2** | **3.24e-2** | **5.89e-2** | 1.24e-1 | 9.26e-2 | **9.67e-2** | **1.08e-1** | **1.38e-1** |
| | PINN | **5.40e-3** | 1.74e-2 | 4.88e-2 | 9.60e-2 | 1.65e-1 | **4.18e-2** | **8.25e-2** | 1.45e-1 | 2.05e-1 | 2.59e-1 |
| Duffing Oscillator | FlowKac | 1.15e-2 | **9.80e-3** | **9.70e-3** | **7.20e-3** | **5.12e-2** | 4.52e-2 | **4.33e-2** | **4.21e-2** | **5.64e-2** | **9.90e-2** |
| | PINN | **5.70e-3** | 1.44e-2 | 3.61e-2 | 7.30e-2 | 1.13e-1 | **2.16e-2** | 5.09e-2 | 9.82e-2 | 1.29e-1 | 1.67e-1 |

Table 2: Comparison of KL divergence and $L^2$ error metrics between FlowKac and PINN models across different SDEs at various time points. Bold values indicate the better-performing model for each metric and time point.

# 6 Conclusion

Neural-based approaches to solving the Fokker-Planck equation have shown remarkable results, particularly when dealing with high-dimensional systems where classical numerical methods suffer from the curse of

| Model | Metric | $t = 0$ | $t = 1$ | $t = 2$ | $t = 3$ |
|-------|--------|---------|---------|---------|---------|
| FlowKac | $D_{\mathrm{KL}}$ | 1.60e-2 | **8.90e-3** | **5.20e-3** | **3.19e-2** |
|         | $L^2$ | 6.07e-2 | **4.70e-2** | **4.74e-2** | **1.37e-1** |
| PINN | $D_{\mathrm{KL}}$ | **1.00e-4** | 4.94e-2 | 2.94e-1 | 1.68 |
|      | $L^2$ | **5.70e-3** | 2.10e-1 | 5.18e-1 | 1.11 |

Table 3: Performance comparison of FlowKac and PINN models on the 2d Ornstein-Uhlenbeck process using KL divergence ($D_{\mathrm{KL}}$) and $L^2$ error metrics.

dimensionality. The widely used Physic-informed approach can encounter convergence challenges as demonstrated through the 2-dimensional Ornstein-Uhlenbeck process.

To address these limitations, we developed FlowKac, a novel framework combining the Feynman-Kac formula and normalizing flows which enables the use of a more robust loss metric improving the stability and accuracy during training. We also developed a stochastic sampling trick that exploits the smoothness of stochastic flows, significantly improving sampling efficiency and reducing computational complexity without sacrificing precision.

Numerical experiments confirm FlowKac's effectiveness, demonstrating strong qualitative and quantitative alignment with true solutions. While higher-order Taylor terms enhance accuracy, they also introduce computational overhead, which must be carefully managed for scalability.

Overall, FlowKac provides a robust and scalable solution for solving the FPE, with promising applications in physics, finance, and other domains requiring accurate modeling of complex stochastic systems.

**Acknowledgments**

We sincerely thank Xiaodong Feng, Li Zeng, and Tao Zhou for providing the code for temporal normalizing flows used in this paper. We also thank Pinak Mandal for helpful discussions.
We thank the anonymous reviewers for their thorough evaluation and insightful recommendations, which helped improve the presentation and clarity of this work.

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

## A  KRnet

The KRnet architecture Tang et al. (2021) comprises two fundamental layers: an Actnorm layer that scales the input and adds a bias, followed by an affine coupling layer responsible for mixing the spatial and temporal variables. The Actnorm layer applies the following transformation:

$$\hat{x}_{[i]} = a_i \odot x_{[i]} + b_i$$

Subsequently, this output is transformed through the affine coupling layer. Let $x_{[i]} = \left[ x_{[i],1}, x_{[i],2} \right]$ be a partition composed of a $m$-uplet $x_{[i],1} \in \mathbb{R}^m$ and $d - m$-uplet and $x_{[i],2} \in \mathbb{R}^{d-m}$, the transformation is expressed as:

$$x_{[i+1],1} = x_{[i],1}$$
$$x_{[i+1],2} = x_{[i],2} \odot \left( 1 + \alpha \tanh \left( \boldsymbol{s_{i+1}} \left( x_{[i],1}, t \right) \right) \right) + e^{\boldsymbol{\beta_i}} \odot \tanh \left( \boldsymbol{r_{i+1}} \left( \boldsymbol{x_{[i],1}}, t \right) \right)$$

where $0 < \alpha < 1$ is a hyperparameter while $\beta_i$ is a trainable parameter. The functions $\boldsymbol{s_{i+1}}$ and $\boldsymbol{r_{i+1}}$ are implemented using neural networks providing the necessary flexibility to capture complex, time-dependent relationships.

To further enhance the nonlinear characteristics of the temporal NF, an additional layer is incorporated and is based on a cumulative distribution function. Consider a probability density function $p(x)$ and a partition of the interval $[0, 1]$ given by $0 = h_0 < h_1 < ... < h_{\hat{m}+1} = 1$. We construct $p(x)$ through a piece-wise linear interpolation:

$$p(x) = \frac{w_{i+1} - w_i}{h_{i+1} - h_i} \left( x - h_i \right) + w_i, \quad \forall x \in [h_i, h_{i+1}]$$

the corresponding cumulative distribution function $F$ is then defined

$$F(x) = \frac{w_{i+1} - w_i}{2(h_{i+1} - h_i)} \left( x - h_i \right)^2 + w_i \left( x - h_i \right) + \sum_{k=0}^{i-1} \frac{w_k + w_{k+1}}{2} (h_{i+1} - h_i), \quad \forall x \in [h_i, h_{i+1}]$$

This layer refines the model's ability to capture and represent complex distributional properties, significantly augmenting the expressiveness and accuracy of the TNF.

## B    Fokker-Planck equation transformation

In this appendix, we provide a detailed derivation of the transformed Fokker-Planck equation. This transformation facilitates the use of the Feynman-Kac formula to derive a solution. The Fokker-Planck equation describing the time evolution of a probability density function $p(x,t)$ for a $d$-dimensional stochastic process with drift vector $\mu = [\mu_1, ..., \mu_d]$ and diffusion matrix $D = \frac{1}{2}\sigma\sigma^\top = [D_{ij}]$ is given by:

$$\partial_t p = -\sum_{i=1}^{d} \partial_{x_i}[\mu_i p] + \sum_{i,j=1}^{d} \partial_{x_i}\left[\partial_{x_j}(D_{ij}p)\right].$$

Expanding the derivative terms in the equation to separate contributions from $p$, $\mu$, and $D$, we have:

$$\partial_t p = -p\sum_{i=1}^{d} \partial_{x_i}\mu_i - \sum_{i=1}^{d} \mu_i\,\partial_{x_i}p + \sum_{i,j=1}^{d} \partial_{x_i}\left(p\,\partial_{x_j}D_{ij} + D_{ij}\,\partial_{x_j}p\right)$$

$$= -p\sum_{i=1}^{d} \partial_{x_i}\mu_i - \sum_{i=1}^{d} \mu_i\partial_{x_i}p + \sum_{i,j=1}^{d}\left(\partial_{x_i}p\,\partial_{x_j}D_{ij} + p\,\partial_{x_ix_j}D_{ij} + \partial_{x_i}D_{ij}\,\partial_{x_j}p + D_{ij}\,\partial_{x_ix_j}p\right)$$

$$= -p\sum_{i=1}^{d} \partial_{x_i}\mu_i + p\sum_{i,j=1}^{d} \partial_{x_ix_j}D_{ij} - \sum_{i=1}^{d} \mu_i\partial_{x_i}p + \sum_{i,j}^{d} D_{ij}\,\partial_{x_ix_j}p + \sum_{i,j}^{d}\left(\partial_{x_i}p\,\partial_{x_j}D_{ij} + \partial_{x_j}p\,\partial_{x_i}D_{ij}\right)$$

Introducing notations for divergence and corresponding second-order operator for compactness, we rewrite the equation:

$$\partial_t p = p\left(-\nabla \cdot \mu + \nabla \cdot (\nabla \cdot D)\right) - \sum_{i=1}^{d} \mu_i\partial_{x_i}p + \sum_{i,j=1}^{d} D_{ij}\partial_{x_ix_j}p + \sum_{i,j=1}^{d}\left(\partial_{x_i}p\,\partial_{x_j}D_{ij} + \partial_{x_j}p\,\partial_{x_i}D_{ij}\right)$$

The symmetry property of the diffusion matrix $D_{ij} = D_{ji}$ further simplifies the last sum:

$$\sum_{i,j=1}^{d}\left(\partial_{x_i}p\,\partial_{x_j}D_{ij} + \partial_{x_j}p\,\partial_{x_i}D_{ij}\right) = 2\sum_{i=1}^{d} \partial_{x_i}p\,\partial_{x_i}D_{ii} + 2\sum_{i\neq j=1}^{d} \partial_{x_i}p\,\partial_{x_j}D_{ij}$$

$$= 2\sum_{i=1}^{d} \partial_{x_i}p\,\partial_{x_i}D_{ii} + 2\sum_{i=1}^{d} \partial_{x_i}p\sum_{j\neq i=1}^{d} \partial_{x_j}D_{ij}$$

$$= 2\sum_{i=1}^{d} \partial_{x_i}p\left(\sum_{j=1}^{d} \partial_{x_j}D_{ij}\right)$$

Combining everything, we obtain a reformulated Fokker-Planck equation with reorganized drift and diffusion contributions:

$$-\partial_t p + \sum_{i=1}^{d}\left(-\mu_i + 2\sum_{j=1}^{d} \partial_{x_j}D_{ij}\right)\partial_{x_i}p + \sum_{i,j=1}^{d} D_{ij}\partial_{x_ix_j}p - \left(\nabla.\mu - \nabla \cdot (\nabla \cdot D)\right)p = 0.$$

## C    Alternating Direction Implicit scheme

We employ the Alternating Direction Implicit (ADI) Papadrakakis et al. (2011) method to solve the 2-dimensional Duffing oscillator. The computational grid is defined with a time step $h = t_{m+1} - t_m$ and a uniform spatial step $\delta = x_{1,i+1} - x_{1,i} = x_{2,i+1} - x_{2,i}$. The governing PDE is given by:

$$\frac{\partial p}{\partial t} = -x_2\frac{\partial p}{\partial x_1} - \left(\omega^2 x_1 - 0.4\omega x_2 - 0.1\omega^2 x_1^3\right)\frac{\partial p}{\partial x_2} + 0.4\omega p + 0.4\frac{\partial^2 p}{\partial x_2^2} \tag{34}$$

The ADI scheme splits the numerical solution into two sequential half-steps, alternating between the spatial dimensions $x_1$ and $x_2$. First, we discretize along the $x_1$-axis using the finite difference method, yielding an intermediate half-step solution $p_{i,j}^{m+\frac{1}{2}}$, denoted as $p_{i,j}^*$. This intermediate solution is then used in the second half-step, where we discretize along the $x_2$-axis to obtain the final time-step solution. The first half-step equation takes the form:

$$\frac{p_{i,j}^* - p_{i,j}^m}{h} = - x_{2,j}\frac{p_{i+1,j}^* - p_{i-1,j}^*}{2\delta} - (\omega^2 x_{1,i} - 0.4\omega x_{2,j} - 0.1\omega^2 x_{1,i}^3)\frac{p_{i,j+1}^m - p_{i,j-1}^m}{2\delta} + 0.4p_{i,j}^m$$
$$+ 0.4\frac{p_{i,j+1}^m - 2p_{i,j}^m + p_{i,j-1}^m}{\delta^2}$$

Rearranging the equation, we obtain:

$$p_{i,j}^* + x_{2,j}\frac{p_{i+1,j}^* - p_{i-1,j}^*}{2\delta}h = (1 + 0.4h)p_{i,j}^m - (\omega^2 x_{1,i} - 0.4\omega x_{2,j} - 0.1\omega^2 x_{1,i}^3)\frac{p_{i,j+1}^m - p_{i,j-1}^m}{2\delta}h$$
$$+ 0.4\frac{p_{i,j+1}^m - 2p_{i,j}^m + p_{i,j-1}^m}{\delta^2}h$$

This equation takes the form of a tridiagonal system:

$$a_i p_{i-1,j}^* + b_i p_{i,j}^* + c_i p_{i+1,j}^* = d_i,$$

where

$$a_i = -\frac{hx_{2,j}}{2\delta}, \quad b_i = 1, \quad c_i = -a_i$$

and

$$d_i = (1 + 0.4h)p_{i,j}^m - (\omega^2 x_{1,i} - 0.4\omega x_{2,j} - 0.1\omega^2 x_{1,i}^3)\frac{p_{i,j+1}^m - p_{i,j-1}^m}{2\delta}h + 0.4\frac{p_{i,j+1}^m - 2p_{i,j}^m + p_{i,j-1}^m}{\delta^2}h$$

We solve this system using the Tridiagonal matrix algorithm (Thomas algorithm) Datta (2010) to obtain the intermediate solution $p^*$.

Afterwards, we apply an explicit finite difference method along the $x_2$-axis. The equation for the next time-step $p^{m+1}$ is given by:

$$\frac{p_{i,j}^{m+1} - p_{i,j}^*}{h} = - x_{2,j}\frac{p_{i+1,j}^* - p_{i-1,j}^*}{2\delta} - (\omega^2 x_{1,i} - 0.4\omega x_{2,j} - 0.1\omega^2 x_{1,i}^3)\frac{p_{i,j+1}^{m+1} - p_{i,j-1}^{m+1}}{2\delta} + 0.4p_{i,j}^*$$
$$+ 0.4\frac{p_{i,j+1}^{m+1} - 2p_{i,j}^{m+1} + p_{i,j-1}^{m+1}}{\delta^2}.$$

This equation is also reformulated into a tridiagonal system:

$$a_i p_{i-1,j}^{m+1} + b_i p_{i,j}^{m+1} + c_i p_{i+1,j}^{m+1} = d_i,$$

where

$$a_i = - \left(\omega^2 x_{1,i} - 0.4\omega x_{2,j} - 0.1\omega^2 x_{1,i}^3\right)\frac{h}{2\delta} - \frac{0.4h}{\delta^2}, \quad c_i = \left(\omega^2 x_{1,i} - 0.4\omega x_{2,j} - 0.1\omega^2 x_{1,i}^3\right)\frac{h}{2\delta} - \frac{0.4h}{\delta^2},$$

$$b_i = \left(1 + \frac{0.8h}{\delta^2}\right)$$

and

$$d_i = (1 + 0.4h)p_{i,j}^* - x_{2,j}\frac{p_{i+1,j}^* - p_{i-1,j}^*}{2\delta}h.$$

The Thomas algorithm is again employed to solve for $p^{m+1}$.

## D    Multivariate GBM closed-form solution

The multivariate GBM process is governed by the SDE:

$$dX_t = \left( A + \frac{1}{2} B^2 \right) X_t dt + B X_t dW_t \tag{35}$$

where $A = \begin{pmatrix} a_1 & 0 \\ 0 & a_2 \end{pmatrix}$, $B = \begin{pmatrix} b_1 & 0 \\ 0 & b_2 \end{pmatrix}$ are diagonal matrices enabling a closed-form solution. The initial condition $\mathbf{X}_0$ follows a log-normal distribution:

$$\psi = \text{Log-}\mathcal{N} \left( \mu_0 = \begin{pmatrix} \mu_{01} \\ \mu_{02} \end{pmatrix}, \Sigma_0 = \begin{pmatrix} \sigma_{01} & 0 \\ 0 & \sigma_{02} \end{pmatrix} \right)$$

The exact solution to the 2D GBM SDE is given by:

$$X_t = \exp\left( tA + BW_t \right) X_0$$
$$= \begin{pmatrix} x_{1,0} \exp(a_1 t + b_1 W_t) \\ x_{2,0} \exp(a_2 t + b_2 W_t) \end{pmatrix}$$

Defining $Z_t = \log X_t = \begin{pmatrix} \log X_{1,t} \\ \log X_{2,t} \end{pmatrix}$, we obtain a Gaussian vector with:

$$\mu_t = \begin{pmatrix} \mu_{01} + a_1 t \\ \mu_{02} + a_2 t \end{pmatrix},$$
$$\Sigma_t = \begin{pmatrix} \sigma_{01}^2 + b_1^2 t & b_1 b_2 t \\ b_1 b_2 t & \sigma_{02}^2 + b_2^2 t \end{pmatrix}.$$

Finally, the probability density function of $X_t$ is:

$$p(x,t) = \frac{1}{x_1 x_2 2\pi \sqrt{\det(\Sigma_t)}} \exp\left( -\frac{1}{2} (\log x - \mu_t)^\top \Sigma_t^{-1} (\log x - \mu_t) \right).$$

# E   Path-wise Jacobian and Hessian computation through Automatic Differentiation

---

**Algorithm 3:** Sampling SDE Solutions and Computing Sensitivities

---

**Input:** SDE with drift $\mu$ and diffusion $\sigma$, time interval $[t_0, t_1]$, number of sample paths $n_{\text{samples}}$, state space dimension $d_{\text{state}}$, number of time points $n_{\text{time}}$, reference point $x_{\text{ref}}$

**Output:** Sample paths $X_t^{x_{\text{ref}}} \in \mathbb{R}^{n_{\text{time}} \times n_{\text{samples}} \times d_{\text{state}}}$, Jacobian $J_t(x_{\text{ref}}) \in \mathbb{R}^{n_{\text{time}} \times n_{\text{samples}} \times d_{\text{state}} \times d_{\text{state}}}$, Hessian $H_t(x_{\text{ref}}) \in \mathbb{R}^{n_{\text{time}} \times n_{\text{samples}} \times d_{\text{state}} \times d_{\text{state}} \times d_{\text{state}}}$

**1 Initialize:** Fix Brownian motion sample paths (`torchsde.BrownianInterval`);

**2 Define function for SDE sampling:**

**3**    expand $x_{\text{ref}}$ to match the batch size $n_{\text{samples}}$;

**4**    Compute SDE sample paths $X_t^{x_{\text{ref}}}$ starting from $x_{\text{ref}}$;

**5 Compute Jacobian:**

**6**    Use automatic differentiation (`torch.func.jacrev`) to compute the Jacobian matrix w.r.t the initial condition $x_{\text{ref}}$;

**7**
$$\text{Jacobian} = \frac{\partial X_t^{x_{\text{ref}}}}{\partial x_{\text{ref}}}$$

**8 Compute Hessian:**

**9**    Define a function that returns the Jacobian for a given input $x_{\text{ref}}$;

**10**    Apply second-order automatic differentiation to compute the Hessian tensor;

**11**
$$\text{Hessian} = \frac{\partial \text{Jacobian}}{\partial x_{\text{ref}}} = \frac{\partial^2 X_t^{x_{\text{ref}}}}{\partial x_{\text{ref}}^2}$$

**12 Return:** Sample paths $X_t^{x_{\text{ref}}}$, Jacobian $J_t(x_{\text{ref}})$, and Hessian $H_t(x_{\text{ref}})$;

---

# F   High-dimensional SDE: quantitative metrics

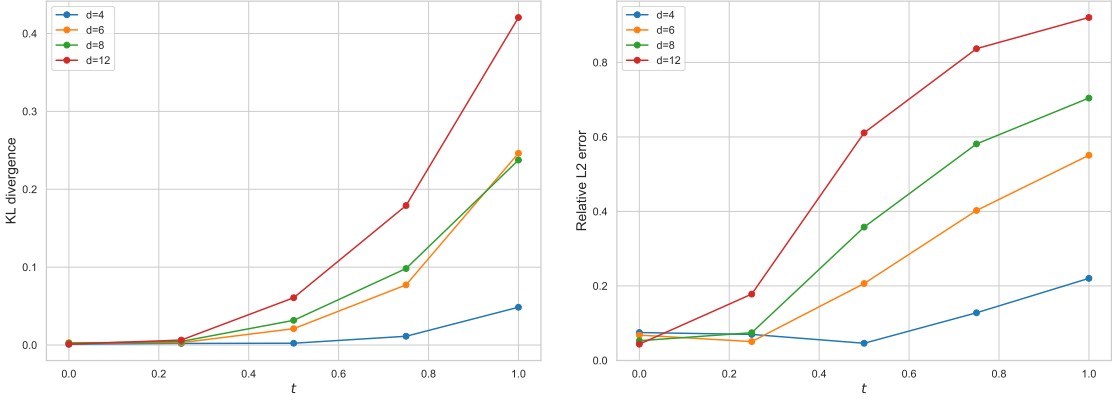

Figure 8: $D_{\text{KL}}$ and relative $L^2$ errors for FlowKac across different dimensional settings (4, 6, 8, 12) and over various time values for the Ornstein-Uhlenbeck process.

Figure 8 presents the relative $L^2$ error and Kullback-Leibler divergence ($D_{\text{KL}}$) across different dimensions ($d = 4, 6, 8, 12$) for the Ornstein-Uhlenbeck process. These results are computed over various time steps and demonstrate the robustness of our approach in accurately modeling high-dimensional stochastic dynamics.

## G    Adaptive sampling

We investigate the role of adaptive sampling, where training points are drawn from the learned density by inverting the normalizing flow, on both training convergence speed and final accuracy. Using the 4-dimensional Ornstein–Uhlenbeck process as a benchmark, we compare training with uniform sampling versus adaptive sampling by tracking the evolution of the relative $L^2$ error and Kullback–Leibler divergence ($D_{\mathrm{KL}}$).

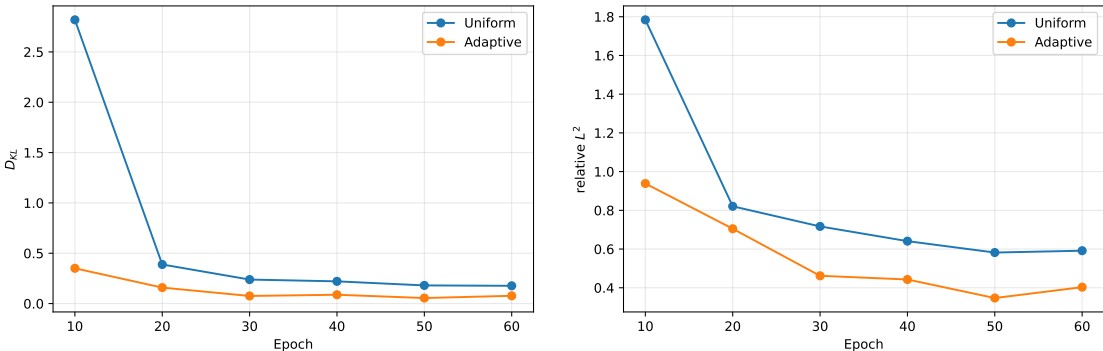

Figure 9: $D_{\mathrm{KL}}$ and relative $L^2$ errors for FlowKac trained with uniform vs. adaptive sampling on the 4D Ornstein–Uhlenbeck process. Adaptive sampling leads to accelerated convergence and improved accuracy.

Although samples are drawn from the learned flow during training, we do not incorporate an explicit importance sampling correction in the loss (Tokdar & Kass, 2010). While a theoretically principled objective would require reweighting by the flow's probability density to account for the change of measure, we find that directly optimizing the empirical loss in equation 4.2 yields more stable convergence and consistently lower errors across experiments.

## H    Ablation study

To better assess the relative influence of the normalizing flow architecture on the model's performance, we conduct an ablation study using the 2-dimensional Geometric Brownian Motion (GBM) process described in subsection 5.3. We vary key architectural and optimization hyperparameters, namely the number of flow layers $L$, the width $W$ of the neural network layers, and the learning rate. The base configuration uses $L = 14$ layers, $W = 32$ neurons per layer, and a learning rate of $10^{-3}$. We report the mean KL divergence and relative $L^2$ errors across time steps $t \in \{0, 0.25, 0.5, 0.75, 1\}$, along with the final training loss, and the best achieved loss to assess convergence behavior.

Table 4: Ablation results on 2D GBM model. Default configuration: `L = 14`, `W = 32`, `lr = 1e−3`. Metrics are averaged across $t \in \{0, 0.25, 0.5, 0.75, 1\}$.

| Configuration | KL Mean ↓ | $L^2$ Mean ↓ | Final loss ↓ | Best Loss ↓ |
|---|---|---|---|---|
| Base ($L = 14, W = 32, \mathtt{lr} = 10^{-3}$) | 4.31e-2 | 1.16e-1 | 7.32e-5 | 4.93e-5 |
| ($\mathtt{Lr} = 10^{-4}$) | 1.37e-1 | 1.67e-1 | 1.85e-4 | 1.78e-4 |
| ($\mathtt{Lr} = 10^{-2}$) | 6.01e-2 | 1.20e-1 | 1.88e-4 | 6.66e-5 |
| ($L = 10$) | 4.32e-2 | 1.25e-1 | 9.23e-5 | 6.86e-5 |
| ($L = 18$) | 2.75e-2 | 9.54e-2 | 4.26e-5 | 3.61e-5 |
| ($W = 16$) | 6.05e-2 | 1.28e-1 | 7.36e-5 | 7.36e-5 |
| ($W = 48$) | 4.42e-2 | 1.16e-1 | 5.45e-5 | 5.45e-5 |

We observe that deeper ($L = 18$) and wider ($W = 48$) architectures yield modest improvements across all metrics. However, these gains are incremental and come at the cost of increased computation and potential overfitting. In contrast, using fewer layers or smaller widths leads to a measurable drop in performance,

but offers better trade-offs in terms of stability and computational efficiency. For this reason, we retain the parsimonious base configuration, which achieves competitive performance across all evaluation metrics while maintaining a manageable training cost.

Interestingly, the learning rate shows a more pronounced effect: both higher and lower learning rates degrade performance, either due to optimization instability or insufficient convergence. This highlights the sensitivity of density estimation in generative SDE modeling to learning dynamics.

## I Runtime comparison

To quantify the practical impact of the stochastic sampling trick, we report a runtime comparison per training epoch between the naive version of FlowKac and the variant leveraging the stochastic sampling trick. The table below summarizes results across various SDEs and dataset sizes. All experiments were run using a batch size of 2000 and 500 Brownian sample paths per point. The results confirm that the stochastic sampling trick offers significant speedups, particularly as the training size and model complexity increase.

Table 5: Runtime comparison (in seconds) between FlowKac (naive) and FlowKac with *Stochastic sampling trick*, for a single training epoch, using a fixed batch size of 2000 and 500 Brownian sample paths. Experiments were conducted on an NVIDIA T4 GPU (16 GB). The speedup factor highlights increasing efficiency for larger training sets and more complex SDEs.

| Configuration | | FlowKac runtime (s) | | Speedup |
|---|---|---|---|---|
| Training Size | Process | Naive | Stochastic Trick | Factor |
| $2 \times 10^4$ | GBM 1d | 31 | **4** | 8× |
| $6 \times 10^4$ | GBM 1d | 93 | **7** | 14× |
| $2 \times 10^4$ | OU 2d | 148 | **5** | 32× |
| $6 \times 10^4$ | OU 2d | 431 | **7** | 61× |
| $2 \times 10^4$ | GBM 2d | 125 | **6** | 20× |
| $6 \times 10^4$ | GBM 2d | 374 | **8** | 47× |

## J Feynman-Kac based MLP comparison

In this appendix, we compare our proposed approach—FlowKac, which combines the Feynman–Kac formula with a temporal normalizing flow and the stochastic sampling trick—with a baseline model that directly approximates the Feynman–Kac and leverages a standard MLP. Both models are evaluated on the 2-dimensional Ornstein–Uhlenbeck process, and we present both quantitative and qualitative results.

**Quantitative Results.** Table 6 reports quantitative error metrics— KL divergence and relative $L^2$ errors—at various time points $t \in \{0; 1; 2; 3\}$. While the Feynman–Kac MLP model performs reasonably well at earlier time points, its accuracy deteriorates over time. In contrast, FlowKac maintains lower errors across all time steps, demonstrating improved stability and fidelity to the target solution.

Table 6: Quantitative comparison between FlowKac and an MLP baseline on the 2D Ornstein–Uhlenbeck process.

| Time $t$ | $D_{\mathbf{KL}}$ (**FlowKac**) | $D_{\mathbf{KL}}$ (**MLP**) | $L^2$ (**FlowKac**) | $L^2$ (**MLP**) |
|---|---|---|---|---|
| 0.0 | 1.60e-2 | 3.38e-2 | 6.07e-2 | 9.53e-2 |
| 1.0 | 8.90e-3 | 3.93e-2 | 4.70e-2 | 9.20e-2 |
| 2.0 | 5.20e-3 | 2.86e-2 | 4.74e-2 | 9.72e-2 |
| 3.0 | 3.19e-2 | 1.11e-1 | 1.37e-1 | 1.91e-1 |

**Qualitative Results.** Figure 10 presents density plots at $t = 2$ and $t = 3$, comparing the outputs of FlowKac, the Feynman–Kac MLP, and the ground truth. While the MLP provides a good approximation, its outputs do not satisfy the properties of a valid probability density function such as non-negativity and proper normalization. In contrast, FlowKac, being a generative model based on normalizing flows, guarantees both by construction.

Furthermore, unlike FlowKac, the MLP does not offer a natural mechanism for sampling, nor can it support adaptive sampling strategies that exploit the learned density during training. This limits its flexibility and effectiveness.

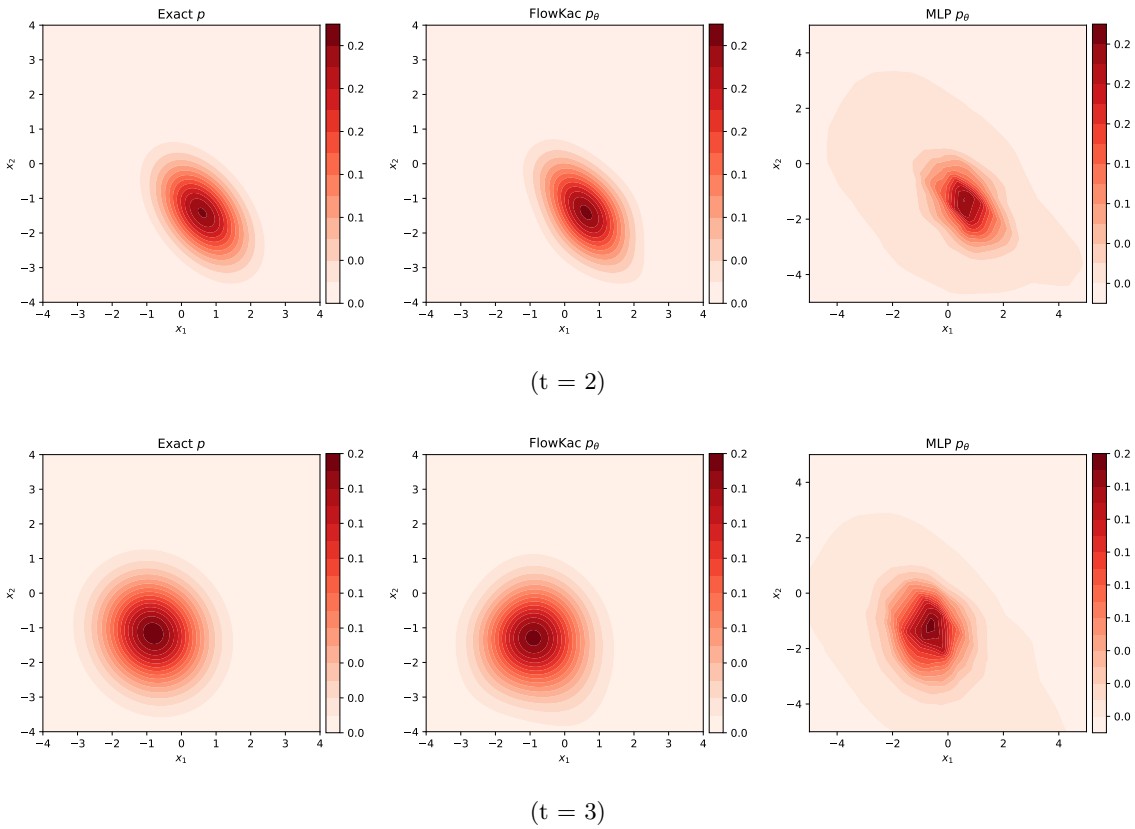

(t = 2)

(t = 3)

Figure 10: Comparison of exact, FlowKac, and MLP densities at times $t = 2$ and $t = 3$.

## K   Alternative dynamic reference point algorithm

As an alternative to using a fixed Taylor expansion reference point $x_{\mathrm{ref}}$, we can dynamically compute it per batch during training. This variant of the FlowKac algorithm enhances accuracy and locality in the Taylor approximation by adjusting the expansion point to match the current batch. The detailed procedure is described in Algorithm 4.

## L   Amortized runtime: Feynman–Kac and TNF

We compare the computational cost of evaluating $n_{\mathrm{eval}} = 10^5$ solution queries, corresponding to 5000 spatial points along 20 temporal points, using two approaches:

- direct application of the Feynman–Kac formula via stochastic sampling, and
- amortized inference with a trained Temporal Normalizing Flow (TNF).

---

**Algorithm 4:** FlowKac (dynamic $x_{\mathrm{ref}}$)

---

**Input** : Maximum epochs $N_e$, number of sample paths $n_W$, number of spatial points $n_x$, number of temporal points $n_t$

**1 for** $l = 1, \ldots, N_e$ **do**

**2**     Sample uniformly $C_{train} = (x^k, t^j)_{1 \le k \le n_x, 1 \le j \le n_t}$;

**3**     Sample $n_W$ Brownian motion paths $W_t^l$ ;       `// Fix realization ω across all points as required by Theorem 4.2`

**4**     Divide $C_{\mathrm{train}}$ into $m$ batches $\{C^b\}_{b=1}^m$;

**5**     Initialize loss $L_e = 0$;

**6**     **for** $b = 1, \ldots, m$ **do**

**7**        Compute $x_{\mathrm{ref}} = \mathrm{centroid}(C^b)$;

**8**        Compute Jacobian $J_{\Phi,t}(x_{\mathrm{ref}})$ and Hessian $H_{\Phi,t}(x_{\mathrm{ref}})$ using automatic differentiation;

**9**        Compute Taylor expansion for batch $C^b$ starting from $x_{\mathrm{ref}}$ (Equation 16);

**10**       Compute $p_{\mathrm{FK}}$;

**11**       Compute batch loss $L_b$;

**12**       Accumulate loss: $L_e = L_e + L_b$;

**13**     **end for**

**14**     Update model parameters $\theta$ using the Adam optimizer;

**15 end for**

**Output:** The predicted solution $p_\theta(x, t)$

---

The results in Table 7 show that the Feynman–Kac incurs an important computational cost. In contrast, once trained, the TNF provides essentially instantaneous evaluations, illustrating the amortization advantage of neural inference.

Table 7: Runtime (in seconds) to evaluate $n_{\mathrm{eval}} = 10^5$ solution queries for two benchmark processes. The Feynman–Kac approach requires sampling at each query, while the TNF offers amortized fast inference.

| Process | Feynman-Kac (s) | TNF (s) |
|---------|-----------------|---------|
| OU 2d   | 50.42           | 0.01    |
| GBM 2d  | 30.32           | 0.01    |

