# OpenReview forum: "FlowKac: An Efficient Neural Fokker-Planck solver using Temporal Normalizing flows and the Feynman-Kac Formula"
_TMLR — Accepted by TMLR_

### Review · Reviewer_Y4Qv · 2025-05-10

**Summary Of Contributions:**

The paper proposes FlowKac, a method for solving the Fokker-Planck PDE for high dimensional complex dynamical systems. It proposes a generative model that combines the Feynman-Kac formula with temporal normalizing flows using physics-informed networks (PINNs).

PINNs, however, have a limitation where minimizing the PINN loss can lead to solutions that do not converge in the case of time-dependent solutions. This limitation is fixed by Feynman-Kac.


Similar to prior work it reformulates the Fokker-Planck equation using Feynman-Kac Feynman-Kac formula which relates PDEs to stochastic processes.
It expresses the solution as an expectation over stochastic paths and sampling such stochastic trajectories. The formula allows a regression loss with the advantage that it allows pointwise evaluation without relying on adjacent points. However, the formula has to be computed for each training point which leads to high computational complexity.

The paper proposes a more efficient way of applying the formula by using the ‘stochastic sampling trick’ with stochastic flows when the flow is k-times differentiable. In that case the flow can be approximated as a Taylor series given the Jacobian and Hessian, and the stochastic paths can be estimated.

It also proposes using temporal normalizing flows as neural sampler and density estimators to adaptively refine the training set so that the procedure concentrates on higher density samples.

As far as I can tell the new contributions of the paper are the stochastic sampling trick and the temporal normalizing flow applied to Fokker-Planck.

Finally the proposed algorithm is evaluated on one and two dimensional systems (univariate and 2d geometric brownian motion, 2d Ornstein-Uhlenbeck process, 2d Duffing oscillator) and higher-order OU process with up to 12 dimensions showing better convergence in the higher dimensional settings.

**Audience:**

Yes

**Claims And Evidence:**

Yes

**Requested Changes:**

Please see the weaknesses above. I think ablation experiments for the normalizing flow component are quite important, since that is a central contribution. Furthermore, please address the questions regarding the flexibility of normalizing flows given the relatively low dimensions.

Please also address the limitations of stochastic sampling, explaining whether the stochastic sampling trick can be used in the cases where a closed form is not known and whether this is a strong limitation.

**Strengths And Weaknesses:**

*Strengths.*

The paper is mostly clearly written and is well-organized with very few errors.
The method is an interesting combination of techniques for stable solution of Fokker-Planck equations.
The paper evaluates the approach on a number of examples, comparing against PINN models.

*Weaknesses:*

What is the significance of the temporal normalizing flow in contrast to a vanilla normalizing flow? What’s the significance of the statement regarding the comparison with neural ODEs and continuous normalizing flows? What is the advantage of maintaining the time invariance?

The modeling power of the normalizing flow depends on the dimension of the input, since each layer has to be a bijection. With low to medium dimensional data this seems like a severe restriction. So even, if one has a complex distribution in, say 3d space, the modeling power of the normalizing flow would be restricted since each layer would have limited width.

What happens if the normalizing flow distribution does not match the true distribution well? Does the adaptive sampling still work or does it correct itself? How much does it depend on a good initial training phase?

Ablation: What is the relative influence of the normalizing flow in the experiments? I don’t think that the authors have made this comparison.

The stochastic sampling trick seems to have some limitations as to applicability. Since it requires the knowledge of the flow at time 0. In the cases where the authors apply this, the closed form solution is always known. The only case where it is not known (the Duffing oscillator), the trick is not applied.

Can you explain the initial training phase? I don’t think this is explained in the paper.


*Typos*
Title: Feynman Kac-Formula -> Feynman-Kac Formula

---

> ### Author Response · Authors · 2025-07-21
> **Response to Reviewer Y4Qv (1/3)**
>
> We thank the reviewer for the insightful review and constructive feedback. We are glad that they found our paper clearly written, well-organised, and appreciated the methodological contributions.
> Below, we address each of the raised points in detail.
>
> ## Comment 1:
> The temporal NF is critical as it explicitly incorporates time into the flow, allowing us to parametrize a time-dependent density $p_t(x)$ through a neural network. This is essential for solving the Fokker–Planck equation, where the target density is inherently time-dependent. In contrast, a vanilla NF does not account for time and only models a *static* density. A naive attempt to model $p_t(x)$ with a vanilla NF by concatenating the spatial variable $x$ and time $t$ into an input $\hat{x} = (x,t)$ leads to incorrect normalization, i.e., $\int p_{vanilla}(\hat{x})d\hat{x} \neq 1$ in general. This behavior is incompatible with the goal of learning a family of densities $\{p_t(x)\}_{t\geq0}$, each normalized over the spatial domain $x$ for fixed $t$, i.e, $\int p_t(x)dx=1 \quad \forall t$.
> The TNF is explicitly designed to mix time without altering the normalization structure and preserving spatial density properties.
>
> Time invariance in the transformation refers to preserving time as an input variable rather than transforming it, which is both mathematically and practically beneficial. Attempting to learn transformations over time introduces complex constraints, such as causality, monotonicity, and positivity, that are difficult to enforce in practice. By contrast, the TNF allows obtaining a new representation $x^{'}$ from both $t$ and $x$, but the new temporal variable $t^{'}$ is unchanged $t^{'}=t$ in order to respect the temporal structure of the problem and avoid unnecessary regularization burdens.
>
> CNFs could also handle time by augmenting the latent space like in (CTFP), but they require solving an ODE for training and inference, which introduces additional computational cost. TNFs sidestep these issues: by parameterizing the density directly as a function of time, we avoid the need for solving differential equations altogether, enabling mesh-free, direct, and efficient density evaluation.
>
> Finally, another conceivable workaround would be to train an independent flow for each time step. However, this not only fails to capture temporal correlations and smoothness across time, but also scales linearly with the number of time steps, making it computationally inefficient and suboptimal.
>
>
> ## Comment 2:
> We acknowledge that the expressiveness of normalizing flows (NFs) can be constrained, particularly due to their bijective architecture. However, our objective is to solve the Fokker-Planck equation by parametrizing a time-dependent probability density $p_\theta(x,t)$ that satisfies three key requirements: positivity, normalization, and boundedness as formalized in Eqs. (3) and (4) in the revised manuscript. Having a generative sampling capability is also a useful feature as it allows sampling from the process with very little cost after the model is trained.
> Normalizing flows are particularly well suited to this task because they inherently enforce the desired constraints by design, and, critically, they provide explicit and tractable density evaluation through the change of variables formula. This is essential for our framework, as the learning objective is framed as density regression, requiring direct access to the model’s density values.
> Moreover, recent empirical evidence has shown that normalizing flows can achieve competitive performance compared to other generative modeling paradigms, including VAEs and diffusion models, across a range of tasks [1].
>
> While other generative paradigms such as diffusion models can model densities, they typically provide access only to the score function, requiring additional steps (e.g., Langevin sampling or Probability flow ODE integration) to obtain samples or densities. Furthermore, these approaches do not naturally represent time-dependent densities, which are central to the solution of the Fokker–Planck equation and central to our pathwise modeling framework.
>
> Alternatively,  normalisation could be manually enforced through grid-based integration, but this reintroduces the very limitations we seek to avoid, namely the curse of dimensionality and the inefficiency of mesh-based solvers.
>
> In summary, despite some architectural constraints, we believe normalizing flows provide an excellent balance between computational tractability and explicit density control, offering an efficient framework for solving the Fokker-Planck equation.

---

> ### Author Response · Authors · 2025-07-21
> **Response to Reviewer Y4Qv (2/3)**
>
> ## Comment 3:
> Without adaptive sampling, training samples are drawn from a uniform distribution, which also does not match the true target distribution. Adaptive sampling accelerates convergence by refining the sampling distribution based on intermediate model outputs, as demonstrated by the L2 errors and KL divergence comparisons for 4D OU processes (please see plots in Appendix G: Adaptive sampling).
>
> Since the normalizing flow is trained simultaneously, its approximation of the true density improves over time, which in turn enhances the quality of the samples used for training. However, adaptive sampling cannot be applied from the very beginning, as the flow is initialised and not trained. In our experiments, we first perform 20 epochs of uniform sampling to ensure a sufficiently accurate initial model before inverting the bijection of the flow and sampling from the flow's density.
> Lastly, although samples are drawn from the normalizing flow, we do not apply an importance sampling correction to the loss, which theoretically ensures the computation of the loss function is unbiased [2]. In principle, when drawing samples from a proposal distribution, the expected value must be reweighted by the inverse of the proposal density to obtain an unbiased estimate. In our case, that would mean dividing the loss by the flow density.
> Empirically, however, we observed that directly using the loss in Eq. (14), without reweighting, yields more stable and accurate results than alternatives based on importance sampling. We have added these precisions to the revised manuscript.
>
> ## Comment 4:
> To evaluate the relative influence of the normalizing flow architecture in our experiments, we conducted an ablation study on the 2D GBM task by varying the number of flow layers $L$, the width $W$ of each layer, and the learning rate. The base configuration uses $L=14$ layers, $W=32$ neurons per layer, and a learning rate of $10^{-3}$.
> We report the summed and averaged KL divergence and relative $L^2$ errors across time steps $t\in\{0, 0.25, 0.5, 0.75, 1\}$, along with the best and final values of the training loss.
>
>
> | Configuration           | KL Sum ↓  | KL Mean ↓| $L^2$ Sum ↓  | $L^2$ Mean ↓ | Final loss ↓ | Best Loss ↓|
> |-------------------------|---------|---------|---------|---------|------------|------------|
> | Base $(L=14, W=32, lr=10^{-3})$                    | 2.15e-1 | 4.31e-2 | 5.81e-1 | 1.16e-1 | 7.32e-5   | 4.93e-5   |
> | Lower lr ($10^{-4}$)    | 6.86e-1 | 1.37e-1 | 8.36e-1 | 1.67e-1 | 1.85e-4   | 1.78e-4   |
> | Higher lr ($10^{-2}$)   | 3.01e-1 | 6.01e-2 | 5.98e-1 | 1.20e-1 | 1.88e-4   | 6.66e-5   |
> | $L=10$                  | 2.16e-1 | 4.32e-2 | 6.27e-1 | 1.25e-1 | 9.23e-5   | 6.86e-5   |
> | $L=18$                  | 1.37e-1 | 2.75e-2 | 4.77e-1 | 9.54e-2 | 4.26e-5   | 3.61e-5   |
> | $W=16$                  | 3.02e-1 | 6.05e-2 | 6.40e-1 | 1.28e-1 | 7.36e-5   | 7.36e-5   |
> | $W=48$                  | 2.21e-1 | 4.42e-2 | 5.80e-1 | 1.16e-1 | 5.45e-5   | 5.45e-5   |
>
> Although configurations with more layers $(𝐿=18)$ or wider networks $(𝑊=48)$ show slightly improved performance, the gains are incremental rather than transformative. To balance performance with computational efficiency and generalization, we opt for the more parsimonious base configuration $(𝐿=14,𝑊=32)$, which already achieves strong results across all metrics. This choice ensures a better trade-off between model complexity and training stability.
>
> We could, in principle, further refine the architecture of the flow to improve results, but this aspect is beyond the scope of the present work, which primarily focuses on the Feynman-Kac approach, the stochastic sampling framework, and its integration with temporal normalizing flows.

---

> ### Author Response · Authors · 2025-07-21
> **Response to Reviewer Y4Qv (3/3)**
>
> ## Comment 5:
> We would like to clarify that the stochastic sampling trick does not require an explicit closed-form solution, nor explicit knowledge of the flow at time 0. Instead, it requires a reference sample path starting from a fixed reference point $x_{ref}$, denoted as $X_t^{x_{ref}}$, which is obtained numerically using standard SDE numerical integration.
>
> The key idea is to leverage this numerically integrated reference trajectory to approximate trajectories starting from other initial points $x$ using a Taylor expansion:
> $$
> X_t^x \approx X_t^{x_{ref}} + \frac{dX_t^{x_{ref}}}{dx_{ref}}(x-x_{ref})+...
> $$
>
> The primary motivation behind this technique is computational efficiency and speedup: it allows us to approximate sample paths from multiple initial conditions without repeatedly invoking costly numerical integration routines. When the dynamics depend linearly or quasi-linearly on the initial condition, higher-order terms are negligible, and the first-order expansion provides exact or almost exact reconstruction.
>
> In contrast, for systems like the Duffing oscillator, where the dynamics are highly nonlinear with respect to the initial condition, we do acknowledge that the sampling trick is limited, as this approximation becomes less accurate. In such regimes, numerical integration from each point is preferable, as it guarantees accuracy even though it is more computationally expensive.
>
> That said, the stochastic trick remains applicable and can still yield reasonable approximations, particularly when the reference point is chosen adaptively (for example, as the centroid of the current mini-batch), although it cannot match the precision of full integration, as some precision is inevitably lost in highly nonlinear regimes.
>
> The reviewer's comment helped us improve the presentation of the stochastic sampling trick. In particular, we replaced the notation $x_0$ with $x_{ref}$ to avoid confusion with initial condition notation. Furthermore, we revised Section 4.3 to better clarify the mechanics of the method.
>
> To sum up, the decision to apply the stochastic trick is guided by a trade-off between efficiency and accuracy. Where its assumptions hold, it offers significant computational savings. Where they do not, full numerical sampling is more appropriate to ensure correctness; this is why we did not apply it in the Duffing case.
>
> ## Comment 6:
>
> The training phase begins by sampling the training set $C_{train}$ from a uniform distribution. We fix the randomness by generating a set of Brownian trajectories, which is crucial to respect the conditions of Theorem 4.2. We then use these Brownian sample paths to compute trajectories of the stochastic process, through numerical integration, starting from a given reference point $x_{ref}$. We also compute the Jacobian matrix of the process with respect to the starting point $x_{ref}$.
> Using the stochastic sampling trick, we then generate trajectories of the process starting from all points in $C_{train}$, which are then used in the Feynman-Kac formula to approximate the solution of the Fokker-Planck equation and compute our loss objective.
>
> We added changes to Algorithm 1 and Algorithm 2 to account for these clarifications.
>
>
> ## References:
> [1] Zhai, Shuangfei, et al. "Normalizing flows are capable generative models." arXiv preprint arXiv:2412.06329 (2024)
>
> [2] Tokdar, S. T., & Kass, R. E. (2010). Importance sampling: a review. Wiley Interdisciplinary Reviews: Computational Statistics, 2(1), 54-60.

---

### Review · Reviewer_9H29 · 2025-07-09

**Summary Of Contributions:**

This paper presents a neural method for solving the Fokker-Planck equation. The neural model used is a normalizing flow augmented with the time dimension, implemented with the KRnet architecture. The ground truth is computed using numerical evaluation of the probability density using the Feynman-Kac formula. The authors also employ a sampling method where new samples are generated in the neighborhood of an existing sample, assuming sufficient regularity. This reduces the sampling complexity of the algorithm. The authors test their method on various stochastic differential equations, and compare their method with physical informed neural network (PINN) solutions.

**Audience:**

Yes

**Claims And Evidence:**

Yes

**Requested Changes:**

* I think the discussion in Sec 4.3 can be improved, and contrasted better wrt. Sec 4.2. For example, the implication of Theorem 4.2 should be explained better with suitable visual representations. And this must be related with the algorithm notations. For example, Algorithm 2 uses batches whereas Algorithm 1 does not.

* Please provide some examples with long term behavior.

**Strengths And Weaknesses:**

Strengths:

* The paper is well written. And the mathematical description is adequate.

* The authors present the algorithm for both the "naive" version as well as the improved version ("stochastic sampling trick"), which helps in the understanding of the key contribution.

* The improved method achieves between 8x and 16x speedup (on at least one example) over the "naive" sampling method.

Weaknesses:

* Since the main contribution of the method is the "stochastic sampling trick", I think, the authors should evaluate the benefit of this method for all the examples. Currently, the only evidence is provided in Table 1 for the univariate Brownian motion example.

* Time interval for all examples is taken as [0,1]. Long term behavior not shown. Will the error be bounded in the long term?

Questions:

* The method is a collocation based method. Are there any restriction from the stability point of view? If yes, how does this method compare against PINN in that regard?

---

> ### Author Response · Authors · 2025-07-21
> **Response to Reviewer 9H29 (1/1)**
>
> We thank the reviewer for the insightful feedback. We are pleased that they found our paper well written, mathematically adequate, and that the presentation of both baseline and improved methods clarified our main contribution. Below, we address each of the reviewer’s comments in detail.
>
> ## Comment 1:
> We thank the reviewer for this suggestion. In response, we have added a new Appendix I presenting a detailed runtime comparison between the naive approach and the version using the stochastic sampling trick, across several SDEs and training sizes. This complements Table 1 and demonstrates the computational benefit of the trick beyond the univariate Brownian motion example.
>
> We note, however, that the runtime comparison does not include the Duffing oscillator. This SDE was modeled using FlowKac with full sampling, as the process exhibits highly nonlinear behavior that limits the applicability of the stochastic sampling trick. In such cases, the approximation is less effective, and precise integration is preferred. This choice is discussed in more detail in our response to Reviewer Y4Qv (comment 5) and Reviewer pZmm (comment 1).
>
> ## Comment 2:
>
> We would like to clarify that the 2d Ornstein-Uhlenbeck (OU) experiment, described in section 5.2, was specifically conducted over an extended time interval [0,3] to illustrate the divergent behavior of the PINN method. This setup allows us to assess the longer-term stability of both methods.
>
> For consistency and comparability across different SDEs, Table 2 presents test metrics restricted to the interval [0,1] as it enables a harmonized benchmark with PINN on a common timescale.
>
> However, to address long-term behavior, we also report additional metrics over the extended interval [0,3] in Table 3. These results provide further evidence of the stability and robustness of our method in longer time horizons.
>
> For longer time periods, we are aware that errors would accumulate, mainly from two sources: the temporal normalizing flow and the numerical integration of the stochastic process, as discretization errors accumulate the farther we sample, for a fixed time step.
>
> However, these limitations are not specific to our approach but are shared more broadly by neural and sampling-based methods. While such behavior is expected in time-dependent models, enhancing long-term accuracy could be explored in future refinements of the framework.
>
> ## Comment 3:
>
> The reviewer raises an important point. While the proposed method is not a classical collocation method in the PDE sense, it does share a collocation-like structure by enforcing the Feynman–Kac identity over a finite set of training points. However, unlike PINNs, it avoids directly enforcing differential constraints and instead relies on stochastic expectations as a solution. Across all our experiments, we did not observe any stability issues with the proposed method.
>
> ## requested changes:
> >* I think the discussion in Sec 4.3 can be improved, and contrasted better wrt. Sec 4.2. For example, the implication of Theorem 4.2 should be explained better with suitable visual representations. And this must be related with the algorithm notations. For example, Algorithm 2 uses batches whereas Algorithm 1 does not.
> >* Please provide some examples with long term behavior.
>
> The implications of Theorem 4.2 are indeed illustrated concretely in Section 5.1 through the example of the 1d geometric Brownian motion (GBM). In this case, visual representations are provided to demonstrate how FlowKac leverages the stochastic sampling trick (as formalized in Theorem 4.2) to approximate the solution efficiently and accurately.
>
> Regarding the distinction between Algorithms 1 and 2, we clarify that the absence of batch notation in Algorithm 1 is intentional: the method requires numerical integration of sample paths individually for each training point $x_k$ from the training set, emphasizing the per-sample cost. In contrast, Algorithm 2 adopts mini-batch notation to reflect the practical speed-up introduced by computing local Taylor expansions around a single reference point $x_{ref}$ (previously denoted as $x_0$, with the change explained in our responses to comment n°1 from Reviewer pZmm and comment n°5 from Reviewer Y4Qv) per batch. This structural difference underlines the practical computational advantage offered by the stochastic trick and connects it directly to the theoretical development in Section 4.3.

---

### Review · Reviewer_31sN · 2025-07-10

**Summary Of Contributions:**

A method for obtaining solutions of the Fokker-Planck equation is proposed. The method fits a (time-dependent) normalizing flow to the solution directly evaluated pointwise using the Feynman-Kac formula. The paper's particular contribution lies mainly in 1) the combination of the use of two notions (normalizing flows and the Feynman-Kac formula), due to which the convergence issue of PINNs for F-P eqs is solved, and 2) the efficient method to compute the Feynman-Kac formula.

**Audience:**

Yes

**Claims And Evidence:**

Yes

**Requested Changes:**

Please consider adding descriptions regarding points (1) and (2) in the Weaknesses section.

**Strengths And Weaknesses:**

### Strengths

- The paper is well written. The related studies and the motivation of the work could be easily followed.
- The method is technically reasonable. Fitting networks directly to the pointwise evaluation of the target solution (because it's available via the F-K formula) makes sense.
- The proposed efficient computation for the F-K formula seems to be effective.

### Weaknesses

(1)
The authors point out that the limitation of the vanilla PINNs for F-P eqs is that a zero PINN loss does not imply the solution is true, referring to Mandal & Apte (2024). I am curious if any empirical results speak of this limitation. Does the inferior performance of the baseline method in Section 5 stem from it? If not, what are the potential causes instead? It seems an important point of discussion because the limitation is the claimed motivation to use the F-K formula for loss definition. I am not sure if the current results support such a claim, or the claim is inherently difficult to support, and thus the superiority of the proposed method should be attributed to other factors too.

(2) In Section 4.2, the authors mention the "dynamic selection of $x_0$," but in Algorithm 2, $x_0$ is simply given as one of the inputs. What is the way to dynamically select $x_0$?

Below are relatively minor points.

(3)
In Theorem 4.1, $q$ seems to appear out of nowhere without any description. It is revealed afterward in the middle of the page, but defining it clearly within the theorem will make it more readable.

(4)
As the pointwise evaluation of the solution is possible by the Feynman-Kac formula, my naive thought is that in practice we could just evaluate it using the proposed efficient computation method, without fitting any networks. I know the point of learning a neural net is amortization, but the need for amortization depends on the computation speed of the non-amortized approach. So I would suggest providing information on the computation time evaluating the solution on $n_\text{eval}$ points, both by the proposed method (neural net) and the mere computation of the F-K formula.

(5) A really minor thing; consider using \eqref or \cref commands to refer to equations.

---

> ### Author Response · Authors · 2025-07-21
> **Response to Reviewer 31sN (1/2)**
>
> We thank the reviewer for their thoughtful and constructive feedback. We are pleased that they found the manuscript well written, clearly motivated, and technically sound. Below, we provide detailed responses to each of the raised concerns.
>
> ## Comment 1:
> We thank the reviewer for raising this important and subtle point. As discussed in Mandal & Apte (2024), a theoretical limitation of the PINN formulation for the Fokker–Planck equation is that minimizing the residual loss does not guarantee convergence to the correct solution; instead, the learned distribution collapses to either the stationary or initial distribution, even as the loss approaches zero.
>
> That said, empirically validating this theoretical failure mode is challenging. While the 2D Ornstein–Uhlenbeck experiment in Section 5.2 shows that the PINN solution significantly deviates from the ground-truth solution over time (despite the loss continuing to decrease), we cannot rigorously assert that this behavior follows the precise mechanism described in Mandal & Apte (2024).
>
> Nevertheless, the empirical evidence aligns with the theoretical concerns: the PINN fails to track the correct time-evolving density despite minimizing its loss. This supports our motivation for adopting the Feynman–Kac-based loss, which guides the model pointwise to the correct target distribution, and thus enforces a stronger (topologically speaking) convergence towards the target.
>
> Finally, we emphasize that both the PINN and FlowKac use the same Normalizing Flow architecture and training settings. Thus, the observed degradation in PINN performance is not due to architectural or optimization differences, but likely due to the loss formulation itself.
>
> ## Comment 2:
>
> In the revised version of the paper, we have clarified this aspect and adopted the notation $x_{\text{ref}}$ (formerly $x_0$) to explicitly denote the reference point used in the Taylor expansion for the stochastic sampling trick.
>
> The dynamic selection of $x_{\text{ref}}$ is implemented as the centroid of the current mini-batch. This strategy is particularly useful when the process is not strictly linear in the initial condition, as it keeps the expansion centered around the data and improves local accuracy. However,  the accuracy of the approximation may degrade in regimes where the dynamics exhibit strong nonlinear dependence on initial conditions, as detailed for the Duffing oscillator.
>
> While the most computationally efficient setting uses a fixed $x_{\text{ref}}$ provided as input (as in Algorithm 2), dynamic selection remains compatible and effective in broader settings.
> Prompted by the reviewers' comments on this matter, we have incorporated this enhancement into the revised manuscript. Specifically, Appendix K and Algorithm 4 have been added to describe the FlowKac variant with dynamic selection of the Taylor expansion reference point.
>
> ## Comment 3:
>
> We agree that the original presentation could be improved for clarity, though there are no mathematical issues with how $q$ appears in Theorem 4.1.
> In the revised version of the manuscript, we have clarified the role of $𝑞$ and positioned it explicitly on the right-hand side of Equation 9 in Theorem 4.1 to highlight its contribution clearly.
>
> More concretely, Theorem 4.1 holds for any function $q \in C(\mathbb{R}^d)$, and for each PDE of the correct form, a corresponding $q$ can be identified. In the case of the Fokker-Planck equation, we rewrite the equation into the form required by the Theorem (as shown in Equation 9), which reveals the corresponding scalar function $q = \sum_{i=1}^d \partial_{x_i}\mu_i -\sum_{i,j=1}^d\partial_{x_ix_j}D_{ij}$.

---

> ### Author Response · Authors · 2025-07-21
> **Response to Reviewer 31sN (2/2)**
>
> ## Comment 4:
>
> It is indeed true that the Feynman–Kac formula allows direct, pointwise computation of the solution via stochastic simulation, without requiring a trained model. However, this approach incurs a significant computational cost as it necessitates simulating multiple stochastic paths per evaluation point in order to estimate the expected value, making it the main bottleneck of the method.
>
> In contrast, once trained, the temporal normalizing flow (TNF) offers extremely fast evaluations at arbitrary test points. The architecture is designed to be highly efficient, and querying the trained model to obtain the solution at any new point incurs negligible computational cost compared to sampling-based evaluations of the Feynman–Kac formula.
>
> While the neural network does require an initial training phase, we emphasize two practical advantages:
> * Amortization: As the reviewer notes, the cost of training is amortized over repeated use, especially when solutions are needed at many new inputs or over different time horizons. This is further improved by the architectural efficiency of the TNF.
> * Adaptive training: The trained TNF can be inverted, thanks to the bijection, and used as a sampler, generating progressively more accurate training data. This adaptivity helps improve both sample quality and training convergence over time.
>
> Below we include a table summarizing the cost of evaluating the solution at $n_\text{eval}=100000$ points, corresponding to $5000$ spatial points along $20$ temporal points, using (i) the Feynman-Kac formula via stochastic sampling, and (ii) the Temporal NF model. This comparison further highlights the computational efficiency gained through amortization and neural inference.
>
> | Process           | Feynman-Kac (s)  | TNF (s)|
> |-------------------|--------------|-----------------|
> | OU 2d | 50.42 | 0.01 |
> | GBM 2d | 30.32 | 0.01 |
>
> The Feynman–Kac evaluation involves repeated stochastic simulations of the underlying SDE to estimate expectations, which constitutes the primary computational bottleneck. In contrast, querying the trained TNF model requires a single forward pass through the neural network, resulting in negligible and constant runtime across all test points.
>
> The higher cost for the OU process relative to the GBM arises from the dimensionality of the driving Brownian motion: the 2d Ornstein-Uhlenbeck process requires simulating two independent Brownian paths per trajectory, whereas the GBM example relies on a shared univariate Wiener process.
>
> This analysis further supports the practical advantage of neural inference: once trained, the model provides efficient and scalable evaluation.
>
> ## Comment 5:
>
> We would like to clarify that we are indeed using the \eqref command for all equation references, in accordance with the journal's formatting guidelines. The rendered appearance is governed by the provided LaTeX template, which controls how references are displayed in the compiled version.

---

> ### Comment · Reviewer_31sN · 2025-08-08
>
> Thank you for the response! It thoroughly answers my questions. The additional explanation for Comment 1 would be helpful for some readers. The runtime comparison given for Comment 4 may be something trivial but nicely supports the need for learning PINNs; giving it somewhere in the appendix or very briefly as a part of the main text (even without the table) would be meaningful.

---

### Review · Reviewer_pZmn · 2025-07-10

**Summary Of Contributions:**

**Summary:** FlowKac introduces a mesh‐free neural solver for the Fokker–Planck equation by recasting it via the Feynman–Kac formula and training a time-conditioned normalizing flow to approximate the evolving density; it leverages stochastic trajectory simulations and an adaptive, generative sampling strategy—accelerated by a local Taylor–expansion trick for Jacobians—to achieve high accuracy and speedups over PINN baselines.

**Audience:**

Yes

**Claims And Evidence:**

Yes

**Requested Changes:**

Please consider addressing my weakness section.

**Minor Comments:** The Codebase link is provided, but the file content is unable to load.

**Strengths And Weaknesses:**

**Strength:**
1) Bypasses expensive grid discretization; complexity grows only with the number of sampled points, not grid resolution.

2) The normalizing flow’s invertibility enables targeted sampling in high-density regions, improving both sample efficiency and final accuracy.

3) Leverages the Feynman–Kac formula for an exact characterization of the solution, rather than approximating PDE residuals.

4) Achieves consistent accuracy improvements( relative l2 error and KL divergence) over PINNs across multiple test SDEs.

**Weakness:**
1) The sampling trick relies on a local expansion around a reference point; for highly nonlinear or multimodal flows, the accuracy of truncated expansions may degrade unless reference points are re-selected continually.

2) Computing higher-order derivatives via automatic differentiation can be memory- and compute-intensive, especially in high dimensions or with deep flow architectures.

3) Comparison is only against PINN-TNF methods; evaluation versus other Feynman–Kac-based neural schemes or classical mesh-free solvers (e.g., FBSDE) would further contextualize gains.

4) Although the method scales beyond 4D, experiments report OOM failures around 12–100 dimensions.

---

> ### Author Response · Authors · 2025-07-21
> **Response to Reviewer pZmm (1/2)**
>
> We thank the reviewer for their thoughtful and constructive feedback. We appreciate their recognition of our method’s sampling efficiency and principled use of the Feynman–Kac formulation. Below, we address each of the reviewer’s comments in detail.
>
> ## Comment 1:
> We thank the reviewer for this valuable observation that helped us improve the presentation of the stochastic sampling trick, by replacing the notation $x_0$ with $x_{ref}$ to avoid confusion with initial notation. To address the limitations of using a fixed reference point in highly nonlinear or multimodal regimes, we have incorporated a mechanism that dynamically selects the reference point at each training iteration. Specifically, the reference point is chosen as the centroid of the current mini-batch. This adaptive strategy ensures that the Taylor expansion is always centered around a point that is representative of the data being processed, thereby improving the local approximation.
>
> That said, for systems where the solution depends linearly (or nearly linearly) on the initial condition, higher-order derivatives beyond the Jacobian vanish. In such cases, a fixed reference point is sufficient, and the stochastic sampling trick provides both accurate approximations and computational efficiency.
>
> These clarifications were added in Section 4.3.
>
> ## Comment 2:
> We agree with the reviewer that computing higher-order derivatives via automatic differentiation can be memory- and compute-intensive, particularly in high dimensions. However, we would like to clarify that the cost of these derivatives depends primarily on the stochastic process itself, not on the complexity of the normalizing flow architecture used to model the density. This is because the derivatives are computed with respect to the SDE trajectories, not the parameters of the NF.
>
> Moreover, the dynamic selection of $x_{ref}$ allows the stochastic sampling trick to rely predominantly on first-order Taylor expansions, which significantly reduces the need for costly higher-order derivative computations in practice.
>
> ## Comment 3:
> We have added a comparison against a Feynman–Kac-based MLP architecture on 2D Ornstein-Uhlenbeck process. To support the evaluation and better contextualize the benefits of our TNF-based approach, we provide both quantitative metrics—KL divergence and $L^2$ errors—and qualitative comparisons in the form of density plots at $t=2$ and $t=3$, where we juxtapose the densities produced by FlowKac, the Feynman–Kac MLP, and the ground truth (see plots in Appendix J: Feynman-Kac based MLP comparison)
>
> | t   | KL (FlowKac) | KL (MLP) | L2 (FlowKac) | L2 (MLP) |
> |-----|--------------|----------|--------------|----------|
> | 0.0 | 1.60E-02     | 3.38E-02 | 6.07E-02     | 9.53E-02 |
> | 1.0 | 8.90E-03     | 3.93E-02 | 4.70E-02     | 9.20E-02 |
> | 2.0 | 5.20E-03     | 2.86E-02 | 4.74E-02     | 9.72E-02 |
> | 3.0 | 3.19E-02     | 1.11E-01 | 1.37E-01     | 1.91E-01 |
>
> These results show that while the Feynman–Kac MLP performs reasonably well at earlier time points, its accuracy degrades over time. In contrast, FlowKac maintains much lower errors. Furthermore, the MLP's output does not always satisfy the properties of a valid density (namely, non-negativity and proper normalization), whereas FlowKac, being a generative model based on normalizing flows, guarantees both by construction.
>
> We note, however, that a direct comparison with the method described in Mandal & Apte (2024) is not feasible in our setting. Their approach relies on modeling or computing the stationary distribution of the Fokker–Planck equation, and combining it with the Feynman–Kac formula to infer the solution at other time values. However, the stochastic processes used in our experiments do not always admit such stationary distributions: the drift matrix $A$ of 2D Ornstein–Uhlenbeck process does not have negative eigenvalues, which is necessary for the existence of a stationary distribution, and has periodic marginal distributions instead, while the 2D GBM collapses into a Dirac mass as $t \rightarrow \infty$.
>
> Finally, due to time and resource constraints, we did not include comparisons against other classes of solvers, although we acknowledge their relevance and plan to explore such evaluations in future work.
>
> ## Comment 4:
>
> We would like to clarify that, as discussed in Section 5.5, out-of-memory (OOM) issues arise only in very high-dimensional settings, specifically at $d \approx 100$, and only when using a single GPU. All reported experiments for dimensions $d\in\{4,6,8,12\}$, as shown in Appendix F, were conducted without encountering any memory limitations. Moreover, the dimensional limit could be extended further through the use of distributed training or multiple GPUs, as well as working on more scalable flow architectures, which we leave for future work.

---

> ### Author Response · Authors · 2025-07-21
> **Response to Reviewer pZmm (2/2)**
>
> >The Codebase link is provided, but the file content is unable to load.
>
> We have re-tested the codebase link and the attached files, and we were able to access and download all contents without any issues. Additionally, the paper includes an anonymous GitHub repository link that provides full access to the codebase. If the reviewer experiences a persistent issue, we would be grateful for more specific details to help us investigate further.

---

> ### Comment · Reviewer_pZmn · 2025-08-12
>
> I thank the author for their detailed response. The additional information will strengthen the manuscript. I also recommend that the author open-source the data and codebase to enhance reproducibility and incorporate the new changes suggested by other reviewers in the final version. Therefore, with new changes, I would recommend accepting the manuscript.

---

### Author Response · Authors · 2025-07-21
**Global response to Reviewers**

We thank the reviewers for their insightful and constructive feedback, which has substantially contributed to improving the quality and clarity of our manuscript. The identified weaknesses and suggestions allowed us to refine the theoretical framework, strengthen the empirical evaluations, and improve the overall presentation of the work.

In response, we have revised the manuscript extensively to address all comments and enhance clarity and robustness. Key improvements include a more detailed theoretical and empirical justification for using Temporal Normalizing Flows (TNFs) over alternative generative models, and the addition of an ablation study to demonstrate the influence of the normalizing flow architecture. We also expanded the experimental comparisons to include another Feynman-Kac-based neural method as an additional baseline.

A major correction involves refining the stochastic sampling trick: we have clarified its assumptions and introduced a dynamic reference point selection to improve the local accuracy of the Taylor expansion. This modification is now formally described in a new appendix.
Additionally, we have added new runtime analysis to quantify the computational efficiency of the stochastic sampling trick.

All the changes to the manuscript are marked in red color, and the list of added appendices is:
* Appendix G: Adaptive sampling
* Appendix H: Ablation study
* Appendix I: Runtime study
* Appendix J: Feynman-Kac-based MLP comparison
* Appendix K: Flowkac with dynamic reference point algorithm

We respond in detail to the points raised by the reviewers below.

---

### Decision · Action_Editor_xGbD · 2025-08-23

**Recommendation:** Accept as is

**Audience:**

Yes

**Audience Explanation:**

This paper could be of interest to researchers studying PDE solvers and PINNs.

**Claims And Evidence:**

Yes

**Claims Explanation:**

The paper introduces FlowKac, a mesh-free neural solver for the Fokker–Planck equation that leverages the Feynman–Kac formula combined with temporal normalizing flows (TNFs) and an adaptive stochastic sampling trick. The efficiency of the method is demonstrated by (i) showing how it circumvents the limitations of PINNs and reduces computational complexity, and (ii) conducting comprehensive experiments on a range of SDEs, in comparison with PINN baselines.